# Membrane protein-regulated networks across human cancers

Chun-Yu Lin [1,16,17], Chia-Hwa Lee[2,3,4,17], Yi-Hsuan Chuang[1], Jung-Yu Lee[1], Yi-Yuan Chiu[1], Yan-Hwa Wu Lee[5,6,7], Yuh-Jyh Jong [5,7,8], Jenn-Kang Hwang[9], Sing-Han Huang [1], Li-Ching Chen[3,10,11], Chih-Hsiung Wu[12,13], Shih-Hsin Tu[3,10,11,12], Yuan-Soon Ho[2,3,14,15] & Jinn-Moon Yang[1,5,7]

Alterations in membrane proteins (MPs) and their regulated pathways have been established as cancer hallmarks and extensively targeted in clinical applications. However, the analysis of MP-interacting proteins and downstream pathways across human malignancies remains challenging. Here, we present a systematically integrated method to generate a resource of cancer membrane protein-regulated networks (CaMPNets), containing 63,746 high-confidence protein–protein interactions (PPIs) for 1962 MPs, using expression profiles from 5922 tumors with overall survival outcomes across 15 human cancers. Comprehensive analysis of CaMPNets links MP partner communities and regulated pathways to provide MP-based gene sets for identifying prognostic biomarkers and druggable targets. For example, we identify CHRNA9 with 12 PPIs (e.g., ERBB2) can be a therapeutic target and find its anti-metastasis agent, bupropion, for treatment in nicotine-induced breast cancer. This resource is a study to systematically integrate MP interactions, genomics, and clinical outcomes for helping illuminate cancer-wide atlas and prognostic landscapes in tumor homo/heterogeneity.

[1] Institute of Bioinformatics and Systems Biology, National Chiao Tung University, Hsinchu 300, Taiwan. [2] School of Medical Laboratory Science and Biotechnology, College of Medical Science and Technology, Taipei Medical University, Taipei 110, Taiwan. [3] TMU Research Center of Cancer Translational Medicine, Taipei Medical University, Taipei 110, Taiwan. [4] Ph.D. Program in Medicine Biotechnology, College of Medical Science and Technology, Taipei Medical University, Taipei 110, Taiwan. [5] Department of Biological Science and Technology, College of Biological Science and Technology, National Chiao Tung University, Hsinchu 300, Taiwan. [6] Institute of Biochemistry and Molecular Biology, School of Life Sciences, National Yang-Ming University, Taipei 112, Taiwan. [7] Center for Intelligent Drug Systems and Smart Bio-devices, National Chiao Tung University, Hsinchu 300, Taiwan. [8] Graduate Institute of Medicine, College of Medicine, Kaohsiung Medical University, Kaohsiung 807, Taiwan. [9] Warshel Institute for Computational Biology, The Chinese University of Hong Kong, 518172 Shenzhen, Guangdong, China. [10] Division of Breast Surgery, Department of Surgery, Taipei Medical University Hospital, Taipei 110, Taiwan. [11] Taipei Cancer Center, Taipei Medical University, Taipei 110, Taiwan. [12] Department of Surgery, School of Medicine, College of Medicine, Taipei Medical University, Taipei 110, Taiwan. [13] Department of Surgery, En Chu Kong Hospital, New Taipei City 237, Taiwan. [14] Graduate Institute of Medical Sciences, College of Medicine, Taipei Medical University, Taipei 110, Taiwan. [15] Department of Laboratory Medicine, Taipei Medical University Hospital, Taipei 110, Taiwan. [16]Present address: Bioinformatics Center, Institute for Chemical Research, Kyoto University, Kyoto 611-0011, Japan. [17]These authors contributed equally: Chun-Yu Lin, Chia-Hwa Lee. Correspondence and requests for materials should be addressed to Y.-S.H. (email: hoyuansn@tmu.edu.tw) or to J.-M.Y. (email: moon@faculty.nctu.edu.tw)

Membrane proteins (MPs) play a key role in mediating intercellular communication and transducing signals in cells through interacting proteins and downstream cellular processes[1]. Alterations in MPs and their regulated pathways, involved in the formation and progression of human cancers[1,2], have been used for development of diagnostic/prognostic biomarkers and pharmaceutical targets[2–4]. Although intensive efforts over the past decades to explore the roles of certain MPs in specific malignancies[2–4], revealing where (tumor type) and how (mechanism) a variety of MPs and their involved pathways contribute to different cancer-associated networks as well as further clinical implications is still a critical challenge.

In support of this pursuit, recent studies have established cell type- and cancer-focused protein–protein interaction (PPI) networks in HEK293T cells[5] and lung cancer cells[6,7], respectively, using large-scale experimental methods; however, these studies are still limited to one cell type (or tumor type) and only one method[7] focuses on MPs and their PPIs. Identification of a set of genes to develop and implement into clinical diagnostic tools is a growing trend[8,9]. Despite a previous work has established the prognostic landscape for individual genes and immune cells across cancers[3], the cancer-wide prognostic landscape of MPs and their regulated pathways (i.e., gene sets) has not been addressed. Therefore, establishing MP-regulated networks across human cancers to illuminate a pan-cancer map and develop clinically applicable molecular models is an unmet need.

Thus, the goal of this study is to simulate regulation patterns of MPs, MP PPIs, and their relevant networks across human cancers, in order to facilitate the development of prognostic stratification and targeted therapy. We first develop a systematically integrated method (SIM) with a scoring system, termed $S_{SIM}$, that identify 63,746 high-confidence PPIs of 1962 MPs. Next, we combine these MPs and their binding partners (i.e., MPP communities) with data from 65 cancer-related pathways[10] and tumor gene expression profiles from 5922 patients[11] to build cancer membrane protein-regulated networks (CaMPNets), including the MP, the MPP community, and MPP community-regulated pathways, for 15 human cancers. Using these CaMP-Nets in conjunction with overall survival data and a meta-analytical framework, we further construct a global pan-cancer landscape to quantify specific/common signatures and prognostic associations in MPP communities and community-regulated pathways. Based on CaMPNets resource (http://campnets.life.nctu.edu.tw), we identify 12 interactions with nicotinic acetylcholine receptor subunit α9 (CHRNA9) across human cancers and validate a use for the Food and Drug Administration (FDA)-approved drug bupropion, which targets CHRNA9, as an anti-metastasis agent in breast cancer. In summary, CaMPNets can reveal the cancer-wide atlas of MPs, MPP communities, and their regulated pathways, with important implications for facilitating the identification of gene set-based prognostic biomarkers as well as therapeutic targets and agents.

## Results

### Identification and analysis of proteins interacting with MPs.
To identify MP-interacting proteins and further establish CaMPNets, we first collected, curated, and integrated three data sets for 2594 MPs (Fig. 1 and Supplementary Data 1). These sets comprised a PPI set containing 749,087 reported PPIs, including 31,810 direct physical PPIs, across 497 species, a pathway set with 292 human pathways, including 65 cancer-related pathways (Supplementary Table 1) from the Kyoto Encyclopedia of Genes and Genomes (KEGG) database[10], and a pan-cancer set from The Cancer Genome Atlas (TCGA) comprising RNA sequencing (RNA-seq) data and clinical outcome data in 15 cancer types

(Supplementary Table 2). Statistical analysis of the reported PPIs and co-expressed gene pairs based on RNA-seq data across the cancer types showed that the numbers of reported PPIs for 2594 MPs were significantly lower than those for the other non-MPs ($P$ value $<3 \times 10^{-16}$, Fisher's exact test; Supplementary Fig. 1), reflecting that many MP-interacting proteins are still unknown.

To identify the interacting proteins of each MP, we proposed a SIM strategy to calculate interacting scores ($S_{SIM}$) by selecting reported PPIs (called reported PPI-based SIM) or direct physical PPIs (called direct PPI-based SIM) as PPI templates (Fig. 1a and Supplementary Fig. 2a–d; details in "Methods"). We sequentially used the potential MP interacting regions (i.e., cytoplasmic regions[12]) to select similar templates by searching reported (or direct) PPIs based on the interacting region similarity ($S_{irs}$) and the quality of the PPI template ($S_{qul}$) (Supplementary Fig. 2b, c). Subsequently, we utilized these selected PPI templates to infer MP-interacting protein candidates by evaluating their $S_{SIM}$ values by searching the complete human proteome database (UniProt[12]; Supplementary Fig. 2d). We compared the PPI prediction accuracies of our $S_{SIM}$, six individual or combined scoring methods[13,14], the STRING database (v. 10.0)[15], the FpClass method[16], and the generalized interolog mapping method[17] (Supplementary Table 3 and Supplementary Note 1) using sets of positive and negative cases (see "Methods" and Supplementary Fig. 3). The result shows that the $S_{SIM}$ approach achieved an average area under the receiver operating characteristic curve (AUC) of 0.924, outperforming those using either source alone (AUC ≤ 0.916), the STRING database (AUC = 0.824), the FpClass method (AUC = 0.811), and the generalized interolog mapping method (AUC = 0.793; Fig. 2a, Supplementary Fig. 4a, and Supplementary Table 4). Similar results were observed for the direct PPI-based $S_{SIM}$ approach (Supplementary Figs. 4 and 5 and Supplementary Note 2). We also examined $F_2$ scores across a broad range of reported PPI- and direct PPI-based $S_{SIM}$ values to determine the threshold for the MP-interacting proteins, and we observed the highest $F_2$ scores of 0.619 and 0.530 when $S_{SIM}$ values were set to 3.6 and 3.7, respectively. To evaluate whether the predictive power was biased toward certain MP types, we classified 2594 MPs into five groups based on the classification/family defined by Almen et al.[1] (Supplementary Data 1). The $S_{SIM}$ scoring method was highly accurate for predicting PPIs of different MP types in comparison to the STRING database, the FpClass method, and the generalized interolog mapping method (Supplementary Table 4). Characterization of the biological functions of the $S_{SIM}$-predicted PPIs (Supplementary Figs. 6 and 7; details in Supplementary Note 3) demonstrated that they displayed high functional similarity, performed/participated in essential properties in humans, highlighted undiscovered regulatory pathways, and were frequently co-expressed in 7208 gene expression sets. Based on the loss-of-function screens of Project Achilles[18] and the mutation and copy-number alteration data of TCGA from the cBioPortal[19] database, we further observed that the percentages of MP PPIs with significant co-occurrence/mutual exclusivity ($P < 0.05$, Fisher's exact test) for our predicted/positive sets in most of the cancer types were significantly higher than those of random chance (empirical $P$ value <0.05; Supplementary Figs. 8 and 9 and Supplementary Note 4). Notably, our results suggest that the gene pairs of the MP-positive/predicted PPIs tend to exhibit co-occurrence of loss-of-function effects (or genomic alterations) but to a limited extent. Finally, we successfully identified 63,746 high-confidence PPIs for 1962 MPs to provide an interactome landscape (Supplementary Data 2), and these data suggest that SIM is a valuable strategy for the discovery of MP PPIs.

We next analyzed the hub properties of MPs and other proteins (called non-MPs) in the PPI networks and investigated

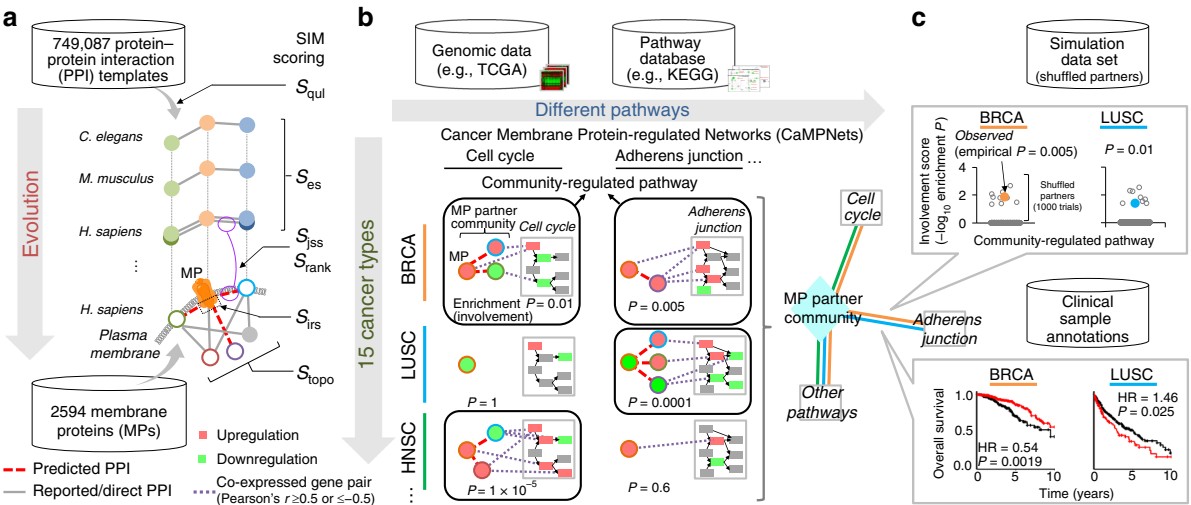

**Fig. 1** Membrane protein (MP) communities and cancer membrane protein-regulated networks (CaMPNets) across human cancers. **a** Schematic description of the systematically integrated method (SIM) for predicting protein–protein interactions (PPIs) of 2594 MPs using a combination of template quality ($S_{qul}$), evolutionary conservation ($S_{es}$), sequence similarity ($S_{jss}$ and $S_{rank}$), interacting region similarity ($S_{irs}$), and network topology ($S_{topo}$). **b** Construction of CaMPNets in human cancers (e.g., breast invasive carcinoma (BRCA), lung squamous cell carcinoma (LUSC), and head and neck squamous cell carcinoma (HNSC)). A CaMPNet consists of an MPP community (an MP with binding partners) and community-regulated pathways. An MPP community-regulated pathway is assessed by the enrichment $P$ values ($\leq 0.05$) of co-expressed gene pairs between the differentially expressed genes (up: red and down: green) of an MPP community and a certain pathway based on gene expression profiles in The Cancer Genome Atlas (TCGA) RNA-seq data. **c** Top, significance (i.e., empirical $P$ value) of the involvement score calculated by 1000 Monte Carlo trials simulated for each community-regulated pathway in individual cancers (for details, see Supplementary Fig. 14a). Bottom, prognostic associations (i.e., adverse or favorable survival) of community-regulated pathways in each cancer type. For Kaplan–Meier curves showing differences in the 10-year overall survival for patients in each community-regulated pathway, an auto-select best cutoff (25–75%) was used, and curve separation was assessed by log-rank test. Clinical outcome data including total 5829 patients in 15 cancer types are assembled from TCGA. HR, hazard ratio

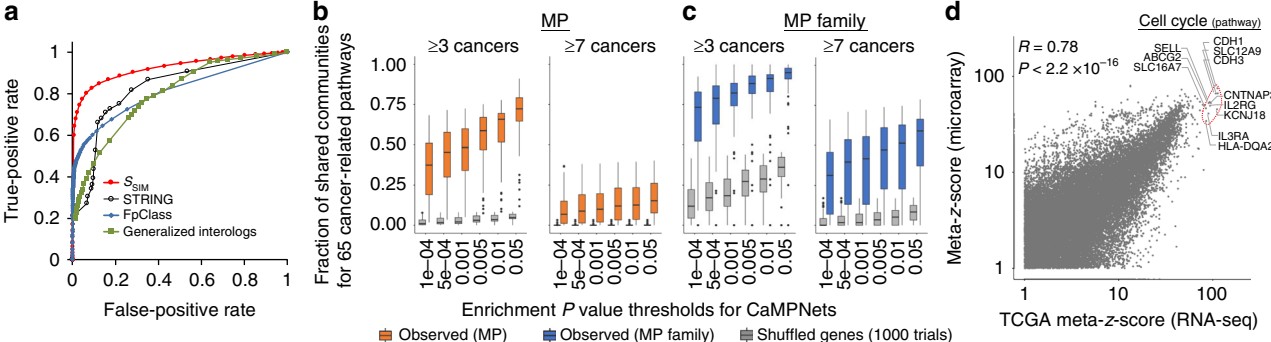

**Fig. 2** Quantification of systematically integrated method (SIM) and MPP (an MP with binding partners) community-regulated pathways in 15 cancer types. **a** Receiver operator characteristic curves of the prediction accuracy of our integrated scoring ($S_{SIM}$) method (red), STRING (black), FpClass (blue), and generalized interologs (green). **b** Analysis of the significance of community-regulated pathways shared by at least three (left) and seven (right) cancers across six enrichment $P$ value thresholds. Among 65 cancer-related pathways, MPP communities were significantly more likely to be involved in a specific pathway in distinct tumor types than was expected by 1000 random trials ($P < 4 \times 10^{-10}$, Wilcoxon signed-rank test). The box represents the interquartile range (IQR) and the horizontal line in the box is the median. The whiskers denote the lowest and highest values within 1.5 times IQR from the first and third quartiles, respectively. The black dots represent outliers. **c** Significance analysis of communities in 197 MP families (blue) for certain pathways shared by multiple cancers ($\geq 3$ or 7 types) across different thresholds of enrichment $P$ values (for details, see Supplementary Figs. 14b and 15). **d** Meta-$z$-scores between The Cancer Genome Atlas RNA-seq data ($n = 5922$ tumors) and 15 independent microarray sets ($n = 528$ tumors) showing a high correlation ($R = 0.78$, $P < 2.2 \times 10^{-16}$, Pearson test). Meta-$z$-scores depict the statistical significance of associations between MPP communities and certain pathways across 15 cancers. Source data are provided as a Source Data file

the topological properties and functional enrichment for the MP-focused and non-MP subnetworks (Supplementary Note 5). In comparison to non-MPs, MPs, which are located mainly in the periphery and not in the center of the cellular interactome, exert limited effects on network integrity (Supplementary Fig. 10) and play roles in cell communication and immune responses on the cell surface (Supplementary Fig. 11). We further characterized the MPP communities by evaluating their compositions and the

overlap between the binding partners identified for different MPs (Supplementary Note 6), suggesting that most communities comprise high percentages of non-MP proteins (Supplementary Fig. 12) and that MPs in a family often share their interacting proteins (Supplementary Fig. 13).

**MPP community-regulated pathways in 15 cancer types.** To build CaMPNets in 15 cancer types, we first identified MPP

community-regulated pathways by evaluating the enrichment $P$ values, measured by hypergeometric distribution, of co-expressed gene pairs of differentially expressed genes (DEGs) between 2594 MPP communities and 65 cancer-related pathways based on TCGA RNA-seq data (see "Methods" and Fig. 1b). Here an MPP community consists of an MP and its interacting partners derived from the reported/direct and predicted PPIs. Next, we determined the empirical $P$ value of each community-regulated pathway based on 1000 permutations by randomly shuffling gene labels of all proteins interacting with 2594 MPs (Fig. 1c and Supplementary Fig. 14a).

First, we used the CaMPNets to quantify tumor homogeneity in community-regulated pathways for MPs (or MP families) across 15 cancers (see "Methods" and Supplementary Fig. 14b). We observed that community-regulated pathways for MPs (or MP families), filtered at enrichment $P$ values ≤0.05 and empirical $P$ values ≤0.05, were significantly more likely to be shared by distinct tumor types than was expected by random chance ($P < 5 \times 10^{-11}$, Wilcoxon signed-rank test; Fig. 2b, c and Supplementary Fig. 15). This result was reproducible across the other statistical thresholds. Communities in MP families (73–95%) involved in 65 cancer-related pathways are shared by multiple cancers (≥3 cancer types) more often than individual MPP communities (37–72%). When comparing individual MPP communities (<16%), ~31–59% of the communities in MP families are connected to certain pathways in at least seven diverse cancers. These results not only are reminiscent of the high cancer-wide concordance reported among genome-wide prognostic genes[3] but also imply MPs in a family often functionally compensate each other to regulate specific pathways in cancers.

To validate the identified MPP community-regulated pathways, we independently tested our CaMPNet approach on an external microarray set containing 19 data sets in 15 cancers from the Gene Expression Omnibus (GEO) database[20] (Supplementary Table 5). In each cancer type, the involvement scores ($-\log_{10}$ enrichment $P$) of community-regulated pathways of both TCGA RNA-seq and microarray data sets showed a significant positive correlation ($R \geq 0.46$, $P < 2.2 \times 10^{-16}$, Pearson correlation, $t$ test; Supplementary Fig. 16). We also observed similar results for the meta-$z$-scores of community-regulated pathways across 15 cancers ($R = 0.78$, $P < 2.2 \times 10^{-16}$, Pearson correlation, $t$ test), reflecting the consistency of using TCGA RNA-seq and microarray data sets (Fig. 2d and Supplementary Fig. 17a). For example, the cell cycle pathway, the most fundamental cancer cell trait for sustaining proliferative signaling[21], was identified as a top one regulated by some MPP communities in both sets. These results show that the construction of CaMPNets is reproducible even using different gene expression resources.

To further investigate whether CaMPNets could identify the undiscovered regulation between MPP communities and cancer-related pathways, we compared the numbers of involved pathways in 15 distinct cancers considering only the MP itself, the MP with reported/direct PPIs, as well as the reported PPI-based and direct PPI-based MPP communities. The MPP communities achieved the highest annotation rate at different thresholds of the co-expressed gene pairs using TCGA RNA-seq data, both with and without filtering by the empirical $P$ value ≤0.05 (Fig. 3a and Supplementary Fig. 18). For example, ~56% of the MPP communities had at least one involved pathway (compared with <31% for MPs themselves and for MPs with reported PPIs). In view of the above results, our strategy is a technique for the comprehensive analysis of MPP community-regulated pathways to reflect their tumor homogeneity and uncover the regulation of cellular processes by MPP communities.

**CaMPNets for pan-cancer analysis**. To investigate the roles of CaMPNets in cancer-wide landscape and cancer hallmarks, we built the CaMPNet-based networks using identified MPP community-regulated pathways. The networks constructed by 1862 reported PPI-based CaMPNets (or 1009 direct PPI-based CaMPNets) in 15 cancers as well as the pan-cancer network (filtered at meta-$z > 1.64$; Supplementary Fig. 17a, b) possessed scale-free network characteristics (Supplementary Fig. 19). In these CaMPNet-based networks, the degree exponent ($\gamma$) values all ranged between 1.184 and 1.990, consistent with the architecture of previously described biological networks[22–24]; smaller $\gamma$ value means that the role of hubs was more important in the network than the network having larger $\gamma$ value[25]. Moreover, we observed that the degree (i.e., regulated pathway number) of the MPP community was proportional to the cancer-wide involvement ($R = 0.78$, $P < 2.2 \times 10^{-16}$, Pearson correlation, $t$ test), which was the mean meta-$z$-score (divided by the degree; Fig. 3b and Supplementary Fig. 17b). This result shows that MPP communities that participate in multiple cellular processes are often involved in many cancers. We therefore considered the communities with degrees within the top 25% of all communities (here, degree ≥26) as the hubs[26] of the pan-cancer network. For example, the amyloid precursor protein (APP) community involved in 58 pathways (i.e., highest degree) across 15 cancers was found to be implicated in common cancer features (Fig. 3b), such as the induction of necroptotic endothelial cell death to promote metastasis and tumor cell proliferation[27,28]. In short, the CaMPNet-based networks display scale-free topology and the MPP community hubs are usually not only found in multiple cancers but also relevant to various cancer hallmarks (Supplementary Note 7).

To obtain a global map of the CaMPNets patterns, we clustered the enrichment-associated meta-$z$-scores of MPP communities across all 65 cancer-related pathways based on hierarchical clustering using the average agglomeration method with correlations as the distance metric (Supplementary Data 3 and 4). We first observed the top-sized clusters for 1862 reported PPI-based MPP communities and 1009 direct PPI-based MPP communities across these cancer-related pathways presenting the similar associations between communities and pathways (Fig. 3c and Supplementary Fig. 20). Among the four largest clusters with >100 reported PPI-based MPP communities, the cluster with 135 communities, such as BDNF/NT-3 growth factor receptor (encoded by *NTRK2*), was broadly linked to the most pathways relevant to cancer hallmarks (Supplementary Fig. 21a and Supplementary Note 7). By contrast, the other three top-sized clusters were relatively specific to certain cancer-related pathways that separately contribute to avoiding immune destruction (Fig. 3d, 279 communities), activating invasion and metastasis (Supplementary Fig. 21b, 119 communities), and evading growth suppressors and virus-induced tumor development (Supplementary Fig. 21c, 105 communities). Our findings also suggest that CaMPNet resource could reflect tissue-specific behaviors of cancers and provide clues to identify common or specific therapeutic targets among different cancer types (Fig. 3d and Supplementary Figs. 21b, c and 22; details in Supplementary Note 7)

We next asked whether our CaMPNets can provide insights into second cancers, defined as histologically distinct cancers that develop in cancer survivors (different from the first cancer). In contrast to other approaches, our CaMPNets achieved better performance, and we found that two cancers with high-profile similarities, not only in the same or adjacent tissues (e.g., COAD and rectum adenocarcinoma (READ)) but also in distinct tissues (e.g., breast invasive carcinoma (BRCA) and uterine corpus endometrial carcinoma (UCEC)), were on the second cancer list

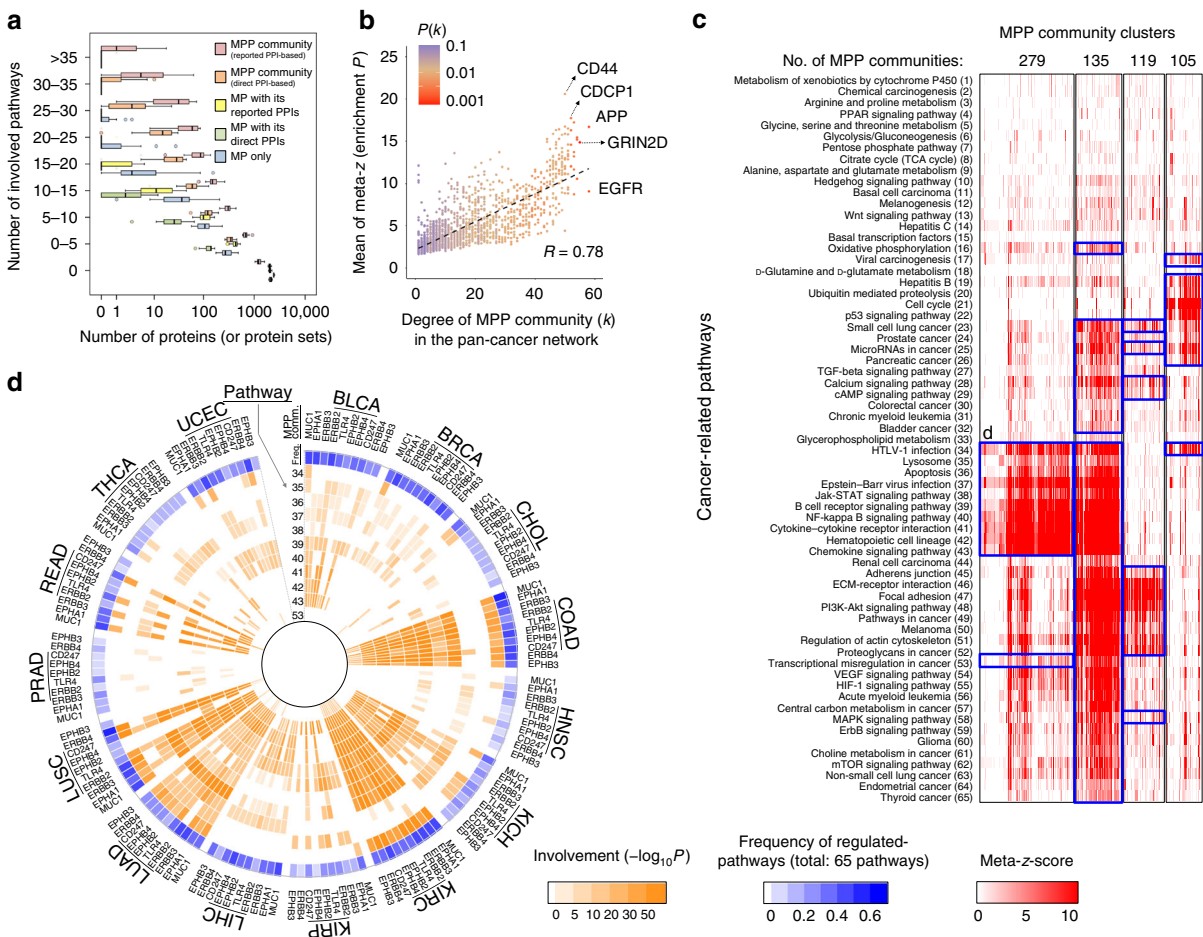

**Fig. 3** Cancer-wide atlas of cancer membrane protein-regulated networks (CaMPNets). **a** Distributions (boxplot) between the numbers of involved pathways and proteins (or protein sets) in membrane protein (MP) itself, MP with reported/direct protein–protein interactions (PPIs), and MP with predicted PPIs and reported/direct PPIs (i.e., reported PPI- and direct PPI-based MPP (an MP with binding partners) communities) across 15 cancers when Pearson's $r$ of co-expressed gene pairs is set to $r \geq 0.5$ or $\leq -0.5$. The box represents the interquartile range (IQR) and the line in the box is the median. The whiskers denote the lowest and highest values within 1.5 times IQR from the first and third quartiles, respectively. Circles and asterisks represent the outliers and extreme outliers, respectively. **b** Scatter plot representation of the regulated pathway numbers (degree $k$) and mean meta-$z$-scores of each MPP community in the pan-cancer network. The probability of the degree, $P(k)$, of each MPP community ranges from low (red) to high (blue). **c** Heat map and hierarchical clustering of meta-$z$-scores for community-regulated pathways between MPP communities and 65 cancer-related pathways in 15 cancers. The four top clusters ranked by cluster size are shown, and details of the clusters for reported PPI-based and direct PPI-based community-regulated pathways are provided in Supplementary Data 3 and 4, respectively. The blue box shows the pathways regulated by most MPP communities ($\geq 50\%$) in each cluster. **d** Involvements (orange) of the top 10 MPP communities with high frequencies of regulated pathways in the largest (first ranked; 279 communities) cluster across human cancers represented in polar histograms; three other clusters are shown in Supplementary Fig. 21. The MPP communities in each cluster are ranked by the frequencies of the regulated pathways (the most external circle) from high (blue) to low (light blue) and zero (white). The internal circles display strong (orange), weak (light orange), and no (white) involvement of community-regulated pathways in the CaMPNet. The number symbols refer to the pathways listed in **c**. Source data are provided as a Source Data file

provided by the American Cancer Society, Inc.[29] (Supplementary Fig. 23; details in Supplementary Note 8). Analogous micro-environments identified by CaMPNets (e.g., MP expression pattern) in different cancers may illustrate the similarities between these cancers as well as the possible causal relationships of first and second cancers.

In brief, CaMPNets could systematically and comprehensively map compositional differences (or similarities) in MPP communities and their regulated pathways across human cancers and be useful for exploring tumor heterogeneity (or homogeneity).

**The prognostic landscape of MPP community-regulated pathways.** To examine whether MPP communities and community-regulated pathways could identify prognostic

associations in cancers, we assessed the association of each MP gene and gene set (i.e., MPP community and community-regulated pathway) with 10-year survival outcomes (see "Methods" and Supplementary Data 5 and 6). Based on the combined scores in 15 distinct cancers, the MPP communities and community-regulated pathways displayed higher frequencies with significant prognostic outcomes ($P < 0.05$, log-rank test) in comparison to MPs themselves, regardless of whether the patients were stratified by the auto-select best cutoff (25–75%)[30] or the median cutoff[3] (50%; Fig. 4a and Supplementary Fig. 24a–c). Similar results for direct PPI-based ones are shown in Supplementary Fig. 25a–d. To further explore cancer-wide prognostic signatures, we used the meta-$z$-scores of adverse and favorable prognostic associations to establish CaMPNets as pan-cancer survival models. The meta-$z$-scores of prognostic

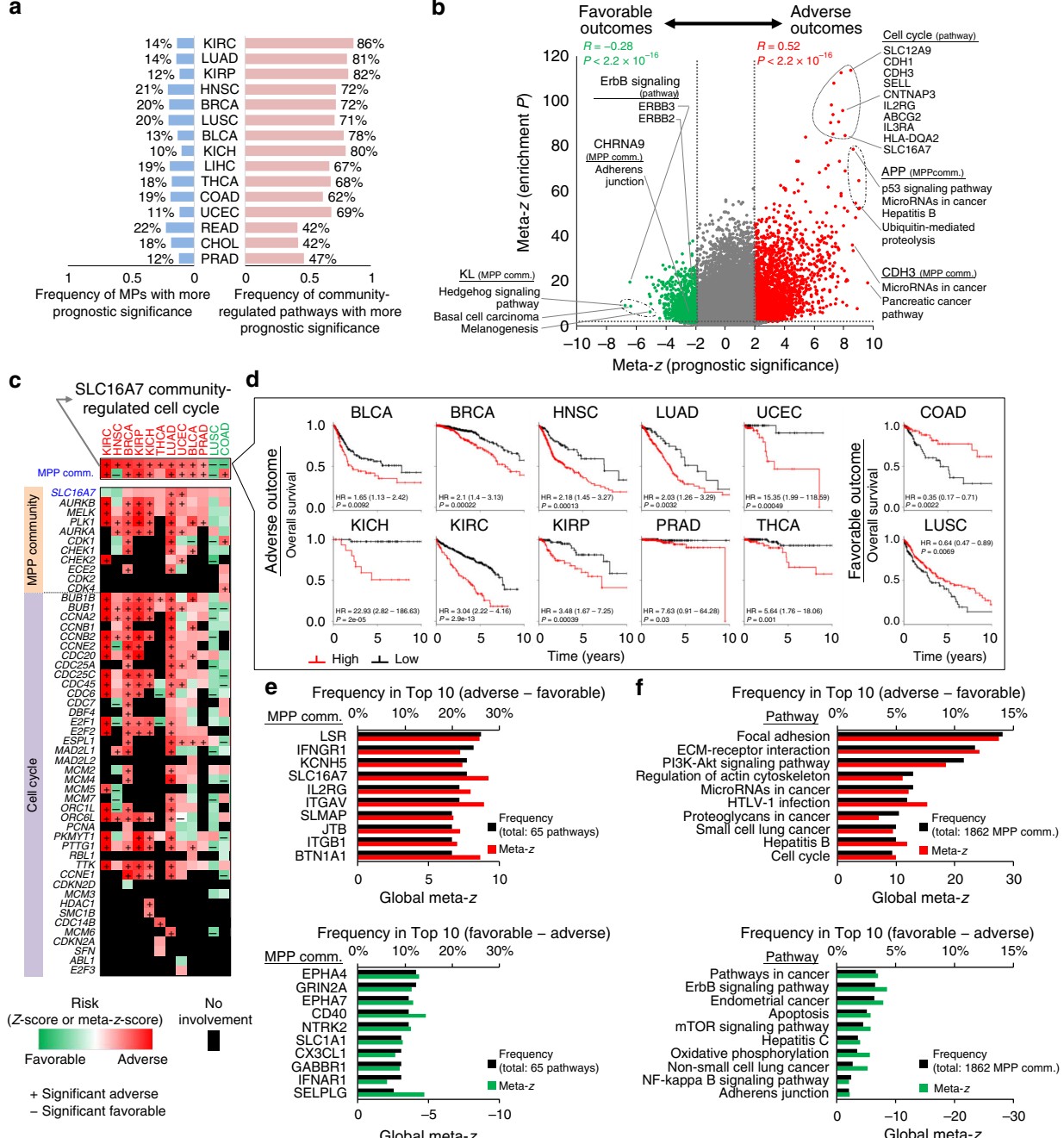

**Fig. 4** Prognostic landscape of cancer membrane protein-regulated networks (CaMPNets) in 15 cancer types. **a** Comparison of prognostic significance between the membrane protein (MP) itself (blue) and the community-regulated pathways (pink) in 15 cancers. **b** Volcano plot of the meta-z-scores of adverse (red; e.g., SLC16A7 community-regulated cell cycle) and favorable (green; e.g., KL community-regulated hedgehog signaling pathway) prognostic community-regulated pathways (x axis) versus the meta-z-scores of the enrichment P values of pathways (y axis) across cancers. The gray dots represent non-enriched community-regulated pathways (meta-z < 1.64 or nominal one-sided P > 0.05) and insignificant prognostic pathways (|meta-z| < 1.96 or nominal two-sided P > 0.05). **c** Heat map of SLC16A7 community-regulated cell cycles in CaMPNets showing survival outcomes of gene members in different cancer types. Statistical significances described by z-scores (or meta-z-score) for associations with adverse prognostic genes (or gene sets) are shown in red, favorable prognostic genes (or gene sets) are shown in green, and genes not involved in the CaMPNets are black. The positive (+) and negative (−) signs indicate significant adverse and favorable prognostic genes (or gene sets) with P < 0.05 (log-rank test). **d** Kaplan–Meier plots representing patients stratified by the auto-select best cutoff (25–75%) for high or low combined scores (see "Methods") of genes in the SLC16A7 community-regulated cell cycle across 12 cancers. Among the 12 cancer types, SLC16A7 community-regulated cell cycles in most cancers have significant adverse outcomes, except for colon adenocarcinoma (COAD) and LUSC, which have favorable outcomes. The 95% confidence intervals are displayed in parentheses. HR, hazard ratio. **e** The ten most frequent cancer-wide adverse (top) and favorable (bottom) prognostic MPP (an MP with binding partners) communities (y axis) considering 65 cancer-related pathways. Notably, the frequencies of pathways for MPP communities with significant adverse and favorable outcomes are subtracted by those with significant favorable and adverse outcomes, respectively. **f** Top 10 frequencies of adverse (top) and favorable (bottom) prognostic pathways (y axis) for 1862 MPP communities in 15 cancers. The global meta-z-score of each MPP community and each pathway reflect 65 pathways and 1862 MPP communities in all cancers, respectively. Source data are provided as a Source Data file

community-regulated pathways and the involvement of these pathways were significantly correlated, especially for positive correlations with adverse prognostic outcomes (red; $R = 0.52$, $P < 2.2 \times 10^{-16}$, Pearson correlation, $t$ test; Fig. 4b). Notably, the number of adverse prognostic community-regulated pathways was higher (2.5-fold, 3046/1209) than those of favorable outcomes, implying that carcinogenicity is relatively common across tumors (as a pan-cancer characteristic), but cancer suppressors may be partially common in specific cancer types (as a component of tumor heterogeneity).

Here we described some of the most significant community-regulated pathways associated with adverse or favorable outcomes as follows and Supplementary Note 9. Regulation of the cell cycle by numerous MPP communities (e.g., SLC16A7, SLC12A9, and CDH1/3) was associated with adverse outcomes and strong involvement (more significant $P$ value). For instance, SLC16A7 that mediates lactate homeostasis in cancer cells, and lactate has emerged as a critical regulator of tumor progression, inflammation, and angiogenesis[31]. In comparison to *SLC16A7* expression being associated with adverse outcomes in only lung adenocarcinoma (LUAD) and UCEC, there is added value in taking into account the genes in the SLC16A7 community-regulated cell cycle for more accurate predictions, thus improving the prognostic power for adverse outcomes in 10 cancers (e.g., kidney renal clear cell carcinoma (KIRC), LUAD, and UCEC) and favorable outcomes in two cancers (i.e., COAD and LUSC; Fig. 4c, d). Of note, the number of genes in the community-regulated pathway demonstrated no correlation with the prognostic association (Supplementary Figs. 24d and 25e). Taken together, these results suggest that MPP communities and their regulated pathways showed more prognostic significance than when considering only the MPs themselves, especially in association with adverse outcomes.

Considering 65 cancer-related pathways (or 1862 MPP communities) across cancers for a specific MPP community (or pathway), we further examined whether a meta-analysis (i.e., global meta-$z$-score; Supplementary Fig. 17c, d) could determine which communities (or pathways) are associated with biological functions required for long-term survival in cancer patients. For example, the SLC16A7 and CD40 communities were the most adverse and favorable prognostic communities, respectively; and the focal adhesion and ErbB signaling pathways were the most adverse and favorable prognostic pathways, respectively (Supplementary Fig. 26; details in Supplementary Note 9). Furthermore, we identified the top ten frequently adverse and favorable prognostic MPP communities (Fig. 4e) and cancer-related pathways (Fig. 4f) relevant to almost all cancer hallmarks[21] (Supplementary Note 9), including promoting tumor inflammation (e.g., IFNGR1 and IL2RG), sustaining proliferative signaling (e.g., PI3K-Akt signaling and cell cycle), evading growth suppressors (e.g., CD40 and apoptosis), and activating invasion and metastasis (e.g., LSR, several integrins, and extracellular matrix–receptor interaction pathway), as well as virus-induced tumor development (e.g., hepatitis B and HTLV-1 infection). Notably, adverse MPP communities and pathways were more likely to be involved in multiple cancers, but favorable communities were relatively discordant (usually in specific cancers) in the engagement of pathways and communities. Therefore, we performed an in silico dissection of CaMPNets to offer routes of access for discovering and developing gene set-based prognostic biomarkers.

**CHRNA9 CaMPNets in cancers.** Nicotinic acetylcholine receptor (nAChR) is a membrane receptor of a neurotransmitter and an ion channel. Several subtypes of nAChR have been indicated

to be closely correlated to the formation of cancers[32–34]. We have previously shown that CHRNA9 is involved in smoking-induced tumor formation in human tumor cells and was highly expressed (mean 7.84-fold) in 186 (67.4%) of the 276 breast cancer paired samples[32,33]. In contrast with several above-mentioned MPs (e.g., CD44, EGFR/ERBB2, and APP) that have been well studied, the interacting partners and pathways associated with CHRNA9 in breast cancer remain to be elucidated.

To further validate our strategy and CaMPNet resource, we generated the CHRNA9 CaMPNets to discover PPIs, to illustrate the cellular functions of CHRNA9 in cancers, to determine whether CHRNA9 community-regulated pathways could be utilized to predict prognosis in patients with different cancer types, and to determine whether existing drugs could be repurposed to target CHRNA9 signaling pathway. First, we identified 64 candidates that could potentially interact with CHRNA9 with $S_{SIM} \geq 3.0$, including 14 candidates with $S_{SIM} \geq 3.6$ and then hierarchically clustered them into 5 subgroups using similarity scores for selecting 18 representative candidates to experimentally validate the method (Supplementary Fig. 27a, b and Supplementary Note 10). Among these 64 candidates, only one (i.e., CHRNA1) has been previously recorded in the STRING database[15] (medium confidence), while none of the other candidates have been previously recorded. In addition, APP[35], EGFR[36], FYN[37–39], and SRC[39–41] have been proposed to bind with other nAChRs (Supplementary Fig. 27c). These suggest that our SIM strategy could identify potential interacting proteins of CHRNA9 and uncover its possible regulated pathways (Supplementary Fig. 27d).

Next, we used our strategy to link the CHRNA9 community, comprising CHRNA9 and its interacting proteins, and 65 cancer-related pathways to construct CaMPNets based on TCGA RNA-seq data in 15 cancer types (Fig. 5a). These CaMPNets in 15 cancers contained 38 total pathways, which were associated with specific cancers (heterogeneity) or up to nine cancers (homogeneity). For instance, our results indicate that the CHRNA9 community was suggested to be implicated in the hepatitis B pathway in five cancers, such as liver hepatocellular carcinoma (LIHC; pink) and KIRC (light blue). Based on the results of microarray, real-time quantitative polymerase chain reaction (Q-PCR), and enzyme-linked immunosorbent assay (ELISA) analyses in Hep3B cells containing an integrated HBV genome in which CHRNA9 was knocked down, we observed that CHRNA9 plays functional roles in the hepatitis B pathway in LIHC, especially in inflammatory-, apoptosis-, and metastasis-related processes (Supplementary Fig. 28 and Supplementary Note 11). Conversely, some CHRNA9 community-regulated pathways, such as cell cycle, adherens junction, and ErbB signaling, were related to cell growth and communication in more than six cancer types, reflecting commonalities across many human malignancies (Fig. 5a). For example, genes of the cell cycle pathway strongly associated with the CHRNA9 and SLC16A7 communities in BRCA and LUAD were significantly altered in MDA-MB-231 and A549 cells in which CHRNA9 and SLC16A7 were knocked down compared to those in the control (Supplementary Fig. 29). These results imply that the CHRNA9 community plays a role in tumor formation, progression, and metastasis.

Our findings also indicated a role for CHRNA9 in breast cancer metastasis. CHRNA9 was found to interact with ERBB2 and EGFR, and their genes are co-expressed with numerous downregulated genes, such as nectin-3 (*NECTIN3*), in the adherens junction pathway to mediate cell–cell adhesion in BRCA (Fig. 5b). Notably, the combined scores of genes in the CHRNA9 community-regulated adherens junction pathway ($P = 0.0019$, log-rank test; Fig. 5c) were significantly associated with favorable outcomes in

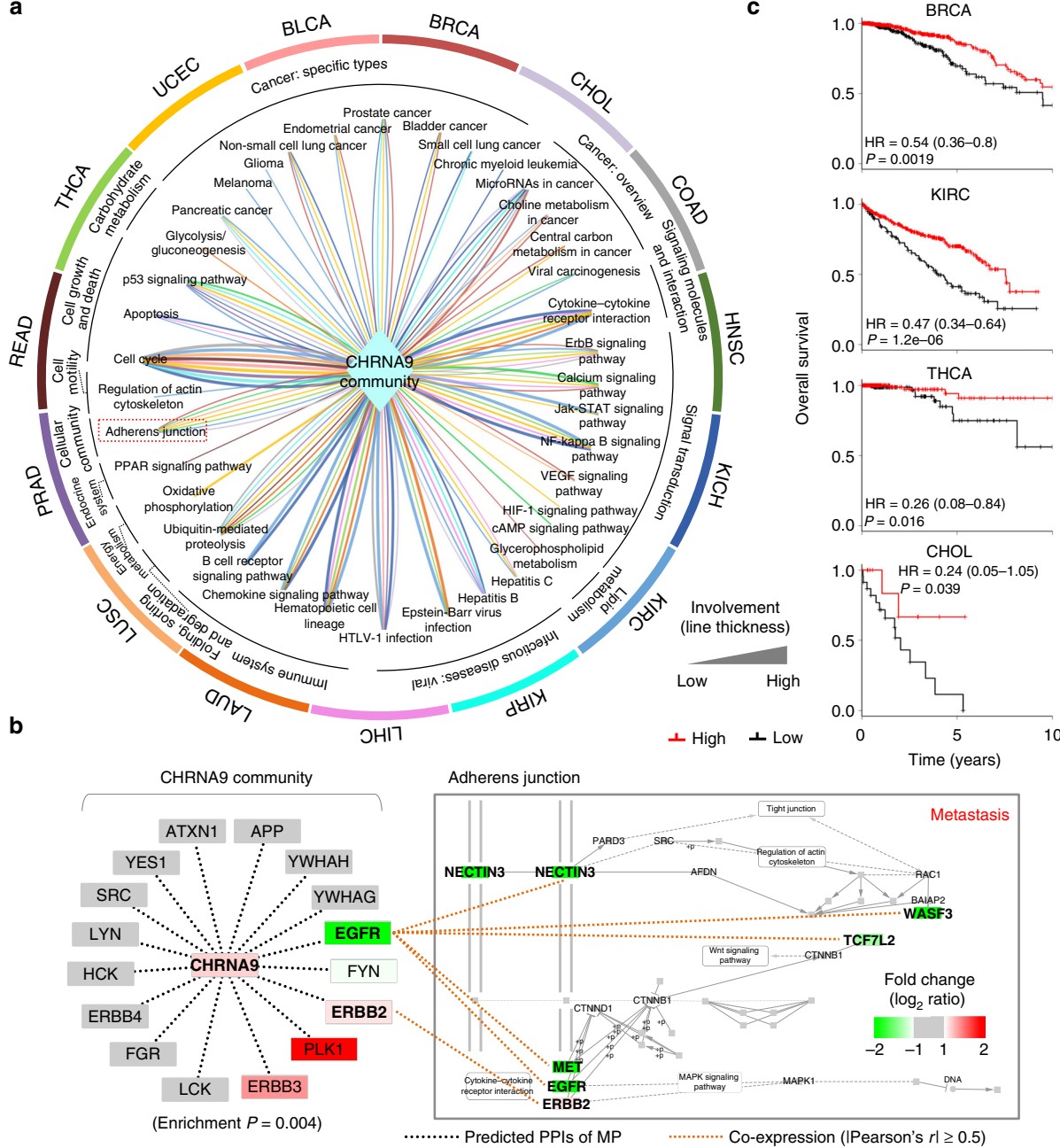

**Fig. 5** Nicotinic acetylcholine receptor subunit α9 (CHRNA9) community-regulated networks in cancers. **a** CHRNA9 cancer membrane protein-regulated networks (CaMPNets) containing 38 cancer-related pathways in 15 cancer types. CHRNA9 and its interacting proteins (i.e., CHRNA9 community) are depicted as a rhombus. These 38 pathways are grouped and labeled according to their Kyoto Encyclopedia of Genes and Genomes categories surrounding the pathway outlines. The colored outlines surrounding the CaMPNets indicate that certain pathways are regulated by the CHRNA9 community in specific cancer types. The thickness of the lines is proportional to the significance of involvement ($-\log_{10}$ enrichment $P$). For example, the CHRNA9 community apparently involves the cell cycle, which is relevant to cell growth and death in most cancers, and participates in adherens junction and vascular endothelial growth factor signaling pathways, which are related to cell adhesion and communication, in BRCA (red) and LUSC (light orange). Source data are provided as a Source Data file. **b** CHRNA9 community-regulated adherens junction in BRCA. The squares with colors and bold letters represent genes in this community-regulated pathway. The black dotted lines denote predicted protein–protein interactions. The orange dotted line indicates a co-expressed gene pair (Pearson's $r \geq 0.5$ or $\leq$–0.5) between genes in the CHRNA9 community and adherens junction. In this pathway, upregulated (red) and downregulated (green) genes, co-expressed with genes in the community, and their interacting neighbors are labeled with gene names. **c** Kaplan–Meier survival plots of differences in 10-year overall survival for BRCA, KIRC, thyroid Carcinoma (THCA), and cholangiocarcinoma (CHOL) patients stratified by the combined score of genes in the CHRNA9 community-regulated adherens junction. Curve separation was assessed by log-rank test

BRCA. These observations suggest that the CHRNA9 community-regulated adherens junction pathway is not only relevant to metastasis but also a significant predictor of favorable survival across several solid tumors (e.g., BRCA and KIRC; Fig. 5c).

**Validation of CaMPNets in cancers**. To experimentally validate our SIM strategy in general, we examined 18 representative interacting partners of CHRNA9 (Supplementary Fig. 27a, b; details in Supplementary Note 10) via immunoprecipitation

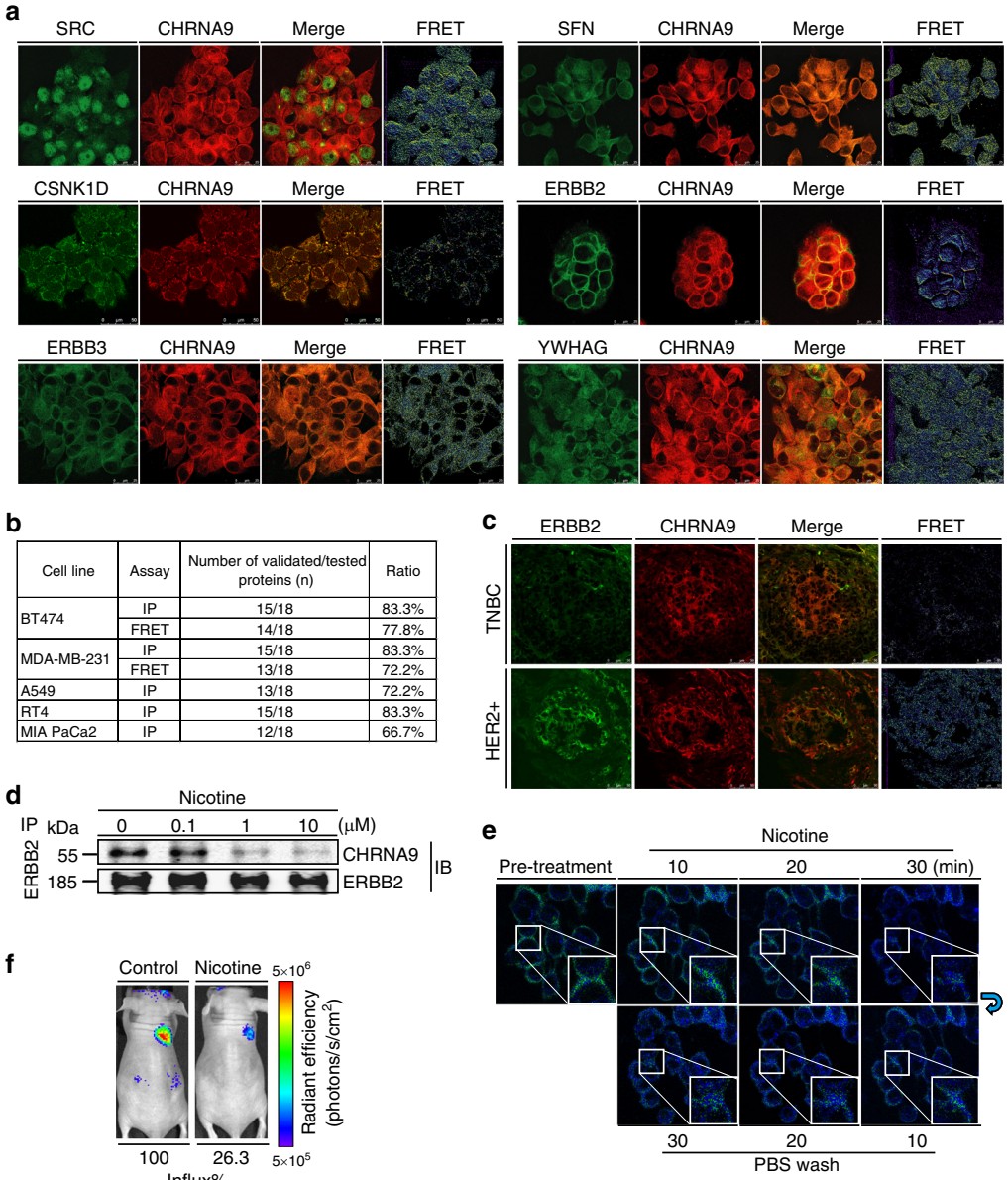

**Fig. 6** Identification of nicotinic acetylcholine receptor subunit α9 (CHRNA9)-binding partners and CHRNA9/ERBB2 interaction changes. **a** Identification of proteins interacting with CHRNA9 by Förster resonance energy transfer (FRET). BT474 cells were hybridized with CHRNA9 and interacting proteins using secondary rhodamine and fluorescein isothiocyanate (FITC) dyes and then examined under a confocal microscope imaging system via FRET module analysis. The representative photos show strong associations between CHRNA9 and the interacting proteins; the results of other partner candidates are shown in Supplementary Fig. 31. The green/yellow/red colors represent the intensity of FRET efficiency. **b** Validation ratios of the partner candidates associated with CHRNA9 via immunoprecipitation (IP) and FRET analyses. **c** The clinical breast tumors of triple-negative breast cancer and HER2-enriched subtypes with CHRNA9/ERBB2 interactions were analyzed by FRET. **d** Dissociation between CHRNA9 and ERBB2 under nicotine dose-dependent treatment. An ERBB2 antibody blotting was used as the loading control. Source data are provided as a Source Data file. **e** Fluorescence-lifetime imaging microscopy (FLIM) of the CHRNA9/ERBB2 complex in MDA-MB-231 cells pretreated with 10 μM nicotine and washed with phosphate-buffered saline. MDA-MB-231 cells were co-transfected with CHRNA9-CFP/ERBB2-YFP 2 days prior to FLIM analysis. The green/blue colors represent the intensity of FRET efficiency. **f** Split luciferase complementation assay of the CHRNA9/ERBB2 complex in a xenograft animal model before and after 100 μg kg$^{-1}$ nicotine oral treatment, as determined by in vivo imaging system detection. MDA-MB-231 cells were co-transfected with CHRNA9-Cluc/ERBB2-Nluc plasmids before being inoculated into the mammary fat pads of nude mice. Luciferase activity was measured before and after 30 min of nicotine administered orally to the same mouse. The color bar indicates a signal gradient from high (red) to low bioluminescence intensity (blue)

(IP; Supplementary Fig. 30) or Förster resonance energy transfer (FRET; Fig. 6a and Supplementary Fig. 31) assays[42] in human cancer cells. The results demonstrated that ≥66.7% of the protein interactions associated with CHRNA9 were identified in BT474 (83.3%), MDA-MB-231 (83.3%), A549 (72.2%), RT4 (83.3%) and MIA PaCa-2 (66.7%) via IP assays; moreover, similar interacting profiles were also discovered in BT474 (77.8%) and MDA-MB-

231 (72.2%) cells by FRET analysis (Fig. 6b and Supplementary Table 6; details in Supplementary Note 12). In addition, we also illustrated that CHK1, CDK1, and PLK1 were associated with SLC16A7 in MDA-MB-231 and A549 cancer cells, providing a 75% validation ratio among four selected candidates (Supplementary Fig. 32). Next, we observed high protein expression of CHRNA9 and ERBB2, having an $S_{SIM} = 3.66$ and displaying a

strong association, in HER2-enriched breast cancer cell lines (e.g., BT474) compared with that in the other cell lines (Supplementary Fig. 33a). Furthermore, a clinical investigation of HER2+ breast tissues revealed a strong protein interaction between CHRNA9 and ERBB2 compared to that in triple-negative breast cancer (TNBC) breast tissues (Fig. 6c). These suggest that our SIM strategy is useful for identifying potential proteins associated with CHRNA9 in various cancer types.

In subsequent experiments, we validated whether CHRNA9 and ERBB2 could form a complex by using nicotine as an agonist for CHRNA9 and investigating downstream signaling in breast cancer cells. We found a significant dissociation of the CHRNA9/ ERBB2 complex with exposure to nicotine at 1 and 10 μM in BT474 cells (Fig. 6d). Our results also suggest that nicotine could induce CHRNA9/ERBB2 interaction change and activate ERBB2 and EGFR downstream signals (Supplementary Fig. 33b–g; details in Supplementary Note 13). To further illustrate the dissociation between CHRNA9 and ERBB2 upon nicotine exposure, we used two-photon confocal microscopy to monitor FRET efficiency with lifetime imaging (i.e., fluorescence-lifetime imaging microscopy (FLIM)) of both fusion proteins[43]. The pretreatment image clearly showed strong FRET efficiency (green) on the cell membrane (Fig. 6e). Following nicotine exposure, the FRET efficiency gradually vanished (blue), and after removing the nicotine by washing with phosphate-buffered saline (PBS), the FRET efficiency on the cell membrane began to recover (turning to green) to baseline, suggesting that nicotine reversibly manipulates interaction between CHRNA9 and ERBB2.

To confirm this finding in an animal model, a split luciferase complementation assay was used to investigate associations and dissociations between CHRNA9 and ERBB2 in MDA-MB-231 cells (Supplementary Fig. 33h–j). We inoculated the mammary pads of nude mice with MDA-MB-231 cells expressing split luciferase fusion proteins and confirmed that the breast tumors expressed high levels of luciferase activity in the tumor regions (Fig. 6f, red–yellow). However, after nicotine exposure, the luciferase activity in the tumor region was dramatically reduced to 26.3% (Fig. 6f, blue) of the original level, indicating that nicotine strongly dissociated CHRNA9/ERBB2 complex formation in this animal model. These observations not only reveal that nicotine exposure activates cell signal transduction of ERBB2 but also imply that CHRNA9 could be a potential therapeutic target in nicotine-induced breast tumorigenesis.

We next asked whether we could prevent dissociation between CHRNA9 and ERBB2 chemically using an existing FDA-approved drug. By using our previous method (Homopharma[44]) and tool (iGEMDOCK[45]), we selected several drug candidates (e.g., fencamfamine, mitotane, and bupropion) as potential CHRNA9 inhibitors (see "Methods" and Supplementary Fig. 34). Among these candidates, bupropion was chosen for use in bioassays because it docked into an allosteric-binding site with the lowest binding energy (Fig. 7a and Supplementary Fig. 35). When tested in vitro, we found that bupropion pretreatment could dramatically inhibit the dissociation of the CHRNA9/ ERBB2 complex with or without nicotine dose-dependent treatment, as determined by an IP assay (Fig. 7b). Bupropion also significantly attenuated nicotine-induced EGFR and ERBB2 phosphorylation in BT474 cells (Supplementary Fig. 36a). Similarly, the inhibitory effect of the CHRNA9/ERBB2 complex disassociation and signal transduction caused by bupropion were also found in lung cancer (A549) cells exposed to nicotine (Supplementary Fig. 37). These results indicate that our screening approach is useful for discovering allosteric-binding inhibitors of CHRNA9.

Based on the finding that the CHRNA9 community-regulated adherens junction pathway is relevant to metastasis, we tested whether bupropion could function as an anti-metastasis agent in breast cancer with and without nicotine stimulation by measuring both the migration and invasion abilities of BT474 and MDA-MB-231 cells. Upon 10 μM nicotine treatment, both BT474 and MDA-MB-231 cells had strong cancer migration (Fig. 7c, d and Supplementary Fig. 36b, c) and invasion (Fig. 7e, f and Supplementary Fig. 36d, e) abilities, whereas pretreatment of these two cell lines with bupropion significantly attenuated their nicotine-induced cancer invasion and migration abilities compared to those of cells treated with the dimethyl sulfoxide control. In addition, cells in which CHRNA9 (or ERBB2) were knocked down showed weak or no changes in their migration and invasion abilities in response to nicotine and bupropion exposure (Supplementary Figs. 38 and 39 and Supplementary Note 14).

Since treatment of TNBC metastasis was considered a challenge in clinic, we further applied bupropion as a nicotine blockade in an MDA-MB-231-based spontaneous pulmonary metastasis animal model[46]. After 2 months of observation, nicotine treatment administered via drinking water significantly increased tumor distant metastasis in lung tissues, as determined by in vivo imaging system (IVIS) imaging (Fig. 7g, up) and photon influx measurements (Fig. 7g, down). Usage of 100 and 200 μg kg$^{-1}$ bupropion three times per week significantly suppressed the number of lung metastasis nodules both with and without nicotine treatment, indicating that bupropion not only blocked the signal from nicotine but also inhibited signals from other endogenous nAChR agonists. Next, we performed a microarray analysis of the above mammary primary tumors to understand the roles of bupropion in anti-metastasis (Supplementary Figs. 40–42; details in Supplementary Note 15). By comparing BRCA tissue samples in TCGA data and treatment with only nicotine, we found that bupropion inhibited signaling cascades in metastasis-related pathways, such as focal adhesion, tight junctions, and adherens junctions, and attenuated nicotine-induced cell metastasis. In summary, the above results indicate that bupropion could suppress metastasis-related pathways.

## Discussion

Because understanding when, where, and how MPs contribute to cancer-focused networks is an emergent need for the development of diagnostic and therapeutic strategies, our CaMPNets represent a resource for delineating MP PPIs and their regulated pathways within and across cancers, illuminating the roles of MPs in tumor homogeneity/heterogeneity and aiding the discovery of gene set-based biomarkers and druggable targets.

CaMPNets have unique advantages over related resources[3,5–7,15,47]. First, the $S_{SIM}$ scoring method systematically simulates MPs undergoing evolutionary processes and environmental forces in time and space. In contrast to the FpClass method, generalized interolog mapping method, and well-known STRING database, our predicted PPIs indeed performed better and added undiscovered MP pathways. Second, multiple RNA-seq and microarray data sets for certain human cancers were included to establish and validate the cancer-focused MPP communities and community-regulated pathways of the CaMPNets. By comparing our data with previous studies focused on one cell[5,47] or tumor type[6,7], this resource further quantified which cancer types and pathways were associated with MPs and which proteins (or genes) of the pathways would interact (or be affected) with these MPs in specific cancers. Third, integrating CaMPNets with a meta-$z$ approach across numerous malignancies provides pan-cancer analysis for revealing which MPs and regulated pathways are specific and which are common tumor hallmarks (Supplementary Note 16). Finally, a set of genes was developed and

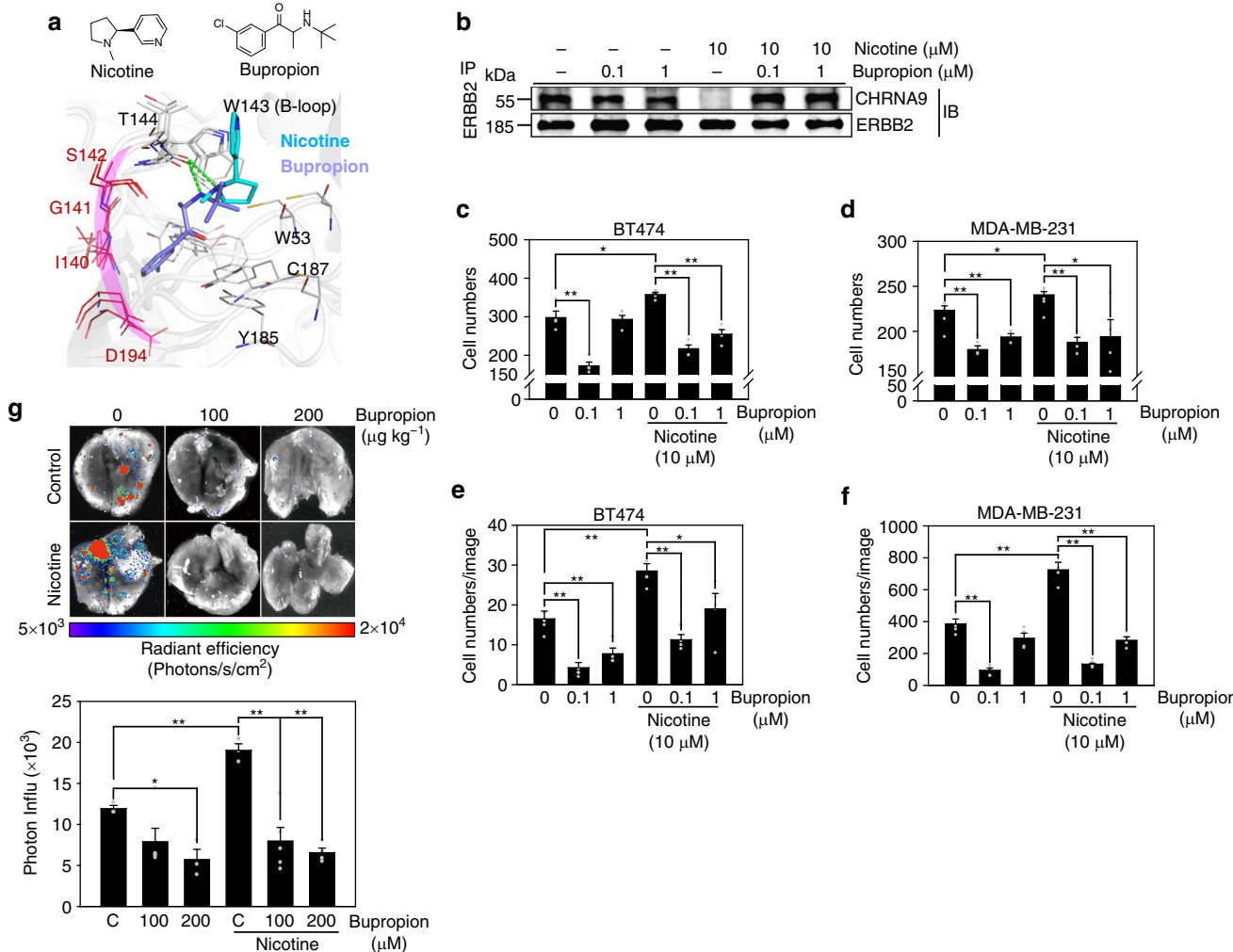

**Fig. 7** Bupropion as a repurposed drug for anti-metastasis in breast cancer. **a** Binding model between nicotinic acetylcholine receptor subunit α9 (CHRNA9), bupropion, and nicotine. Compared with nicotine, the 1-(3-chlorophenyl)propan-1-one bupropion moiety can bind an additional subpocket comprising several CHRNA9 contact residues (e.g., I140, G141, S142, and D194). **b** Bupropion as an inhibitor attenuating nicotine-induced CHRNA9/ERBB2 dissociation, as detected by an immunoprecipitation (IP) assay. BT474 cells exposed with or without 10 μM nicotine were subjected to bupropion pretreatment at 0, 0.1, and 1 μM. An ERBB2 antibody/bead complex was precipitated and immunoblotted for a CHRNA9 antibody, and an ERBB2 blotting was used as the loading control. Bupropion as a repurposed drug not only inhibits the **c**, **d** migration and **e**, **f** invasion abilities of BT474 (24 and 72 h) and MDA-MB-231 (12 and 48 h) cells, and **g** the metastasis ability of a spontaneous pulmonary metastasis mouse model but also attenuates the nicotine-induced effects. In invasion and migration assays, the cells were either treated with or without nicotine (10 μM). All mice were randomized into six groups (*n* = 5 per group) and intraperitoneally injected with phosphate-buffered saline, 100 or 200 μg kg$^{-1}$ bupropion three times per week, and with or without nicotine treatment (10 μg ml$^{-1}$) via their drinking water. After 2 months, the mice were sacrificed, and their lung tissues were measured for metastasis by bioluminescence imaging. The color bar indicates a signal gradient from high (red) to low bioluminescence intensity (blue). The error bars indicate the mean ± standard error. Data were analyzed with Student's *t* tests; all *P* values were two-sided. *P* values <0.05 and <0.01 are indicated by an asterisk and double asterisk, respectively. Source data are provided as a Source Data file

implemented into clinical diagnostic tools, such as Mamma-Print[8] and Oncotype DX[9]. CaMPNets offer a framework to guide the design of prognostic gene set tests via MPP communities and their regulated pathways across human cancers. This resource links MP interactions, genomics, and clinical outcomes to inform biological and diagnostic/therapeutic strategies.

Our resource also offers clues for observing how changes in cancer-related pathways or MPP communities reflect clinical outcomes, such as alterations in prognostic associations (details in Supplementary Note 16). In addition, our observations may explain why multi-target therapeutics are effective and overcome adaptive resistance to cancer therapy[48] since MPs belonging to

the same family often display complementary functions toward each other in mediating certain pathways in distinct human cancers. This study also suggests that bupropion could have utility in smoking-related metastatic cancer patients with high nAChR expression (Supplementary Note 16). However, the target proteins (e.g., nAChRs) of bupropion to suppress nicotine-induced complex disassociation, downstream signals, and metastatic ability in BRCA remain to be fully elucidated. Even so, the integration of CaMPNets and homopharma will be useful for the future development of precision medicine.

CaMPNets have several limitations, challenges, and perspectives (details in Supplementary Note 17). First, the predicted PPIs identified by reported PPI- or direct PPI-based SIM still need to

be experimentally validated. Second, one potential limitation of CaMPNets is that our approach may miss gene sets that belong to the same pathway but are potentially not sufficiently co-expressed, as the co-expression may be less evident in the case signaling pathways that are often hierarchical in nature. Third, we believe that our approach is a general strategy for identifying interactions of other MPs and further constructing disease-associated networks via corresponding genomic data. Fourth, another CaMPNets challenge is to consider interactions between MPs and extracellular proteins for elucidating tumor microenvironment responses. The unabated progress in single-cell sequencing and next-generation sequencing technologies[49] will allow this issue to be addressed as well as revolutionize our model to reconstruct close-to-real CaMPNets.

In conclusion, our results shed light on the cancer-wide atlas and prognostic landscape of both MPP communities and their regulated pathways, as well as providing numerous clues for further investigation and clinical translation. Our resource also promotes discoveries of MPs and their PPIs as promising targets for the development of biomarkers and therapeutic targets. According to our knowledge, our resource and approaches provide a useful framework to facilitate the discovery of MP PPIs, PPI modulators, communities, regulated pathways, and further clinical applications.

## Methods

**MP set.** MPs can be broadly classified into integral (intrinsic) and peripheral (extrinsic) proteins according to the nature of their membrane–protein interactions. Here we focused on integral MPs in the plasma membrane, which are intrinsic to the plasma membrane and contain transmembrane region(s) as well as extracellular and/or cytoplasmic region(s). The cytoplasmic region of an integral MP plays a key role in conveying signals into cells by interacting with other proteins, including direct binding and phosphorylation in intracellular signaling pathways. We restricted our focus to identify the interacting proteins of MPs in the plasma membrane within tumor cells to comprehensively establish intracellular CaMPNets across human cancers. Therefore, we selected 2594 MPs from the UniProt complete proteome database[12] based on the following criteria (Supplementary Data 1): an MP is annotated with (1) the specific "plasma membrane" term or its children's terms (e.g., plasma membrane receptor complex and integral component of plasma membrane) of cellular components (CCs) in the gene ontology (GO) database and (2) "Cytoplasmic" of topological domains as well as either "Transmembrane" or "Intramembrane" in the topology feature of the UniProt Knowledgebase. To further characterize the 2594 MPs, they were classified into 214 families belonging to 5 functional groups, including receptors (1073 members, 1073/2594 = 41.4%), transporters (450, 17.3%), miscellaneous (296, 11.4%), enzymes (89, 3.4%), and unclassified (686, 26.4%), according to the study by Almén et al.[1].

**PPI data sets.** To predict proteins that interact with the MPs, we first collected 749,087 reported and non-redundant PPIs across 497 species (called the reported PPI set; Supplementary Fig. 43) from 5 public databases, including IntAct[50], BioGRID[51], DIP[52], MIPS[53], and MINT[54], and then filtered by considering both two proteins of each PPI recorded in the UniProt complete proteome database. The information that was collected and processed during curation of these PPIs included the UniProt accession numbers, gene names, and species for both proteins of the interaction; the associated PubMed identifier; and the identifier number and name of the interaction element following the standard "interaction detection method" and "interaction type" vocabulary implemented in the Molecular Interactions (MI) of HUPO Proteomics Standards Initiative (PSI)[55]. To further identify direct physical interactions from 749,087 reported PPIs, we first used the term "experimental interaction detection (MI:0045)," which indicates the methods based on laboratory experiments to determine an interaction, and its subclass terms to filter the reported PPIs that have no MI annotation term related to experimental validation. Based on the definition of a direct physical interaction in the DIP[52], IntAct[50], and PICKLE[56] databases, we then used the term "direct interaction (MI:0407)," which is defined as an interaction between molecules that are in direct contact with each other, and the subclass terms of direct interaction (e.g., covalent binding, MI:0195) to determine whether a reported PPI was a direct interaction candidate. According to criteria 1 and 2, 174,193 reported PPIs were classified as direct interaction candidates.

Next, we assigned the scores to experimental interaction detection methods according to the reliability of experimental techniques defined by HIPPIE[57] and BioGRID[51]. The scores ranged from 0 (lowest confidence) to 10 (highest confidence); for example, X-ray crystallography and genetic interference were given

the highest and lowest scores, respectively. In the BioGRID database, these two techniques were also separately deemed to be approaches for detecting the direct and physical interactions and the synthetic/suppressive/additive genetic interactions defined by inequality. In addition, the high-throughput screening experiments, such as the two-hybrid screening and the mass spectrometry-based proteomics, were assigned scores ≤5. If an MI term lacked an assigned score, it inherited the score from the nearest parental term. Moreover, the child term was chosen as the representative term and its score was used when one term belonged to the subclass term of another one. To avoid high-throughput screening experiments lacking secondary experimental validation, we only selected the direct PPI candidates with a sum of scores ≥6. Finally, 749,087 reported PPIs (or 31,810 direct PPIs; Supplementary Data 7) were used as PPI templates to predict the MP-interacting proteins; notably, to avoid bias in evaluating the predictive power of our method, all reported PPIs (or direct PPIs) of each MP were excluded in advance of the PPI templates being selected by sequence alignment of the cytoplasmic region of that specific MP.

To evaluate the reliability of the predicted PPIs for 2,594 human MPs derived from our method, we further curated three data sets, two standard positive (SP) sets and a negative (SN) set. The positive cases of MPs in two SP sets consisted of 18,827 reported PPIs and 2049 direct PPIs in humans derived from the reported PPI set. In the SN set, the negative protein pairs were defined using relative specificity similarities ($RSS_{BP}$ and $RSS_{CC}$) between GO biological processes (BPs) and GO CCs, as proposed by Wu et al.[58] (Supplementary Fig. 3). Our results showed that >95% of the human PPIs (or PPIs of MPs) in the reported PPI set had $RSS_{BP} \geq 0.4$ or $RSS_{CC} \geq 0.4$. Here 555,438 and 75,799 protein pairs, for which $RSS_{BP} < 0.4$ or $RSS_{CC} < 0.4$, were considered negative cases among 4,500,936 and 774,751 candidates with joint sequence similarities (joint $E$ value) ≤ $10^{-40}$ based on 749,087 PPI templates and 31,810 direct PPI templates, respectively. To further evaluate the essentiality of the PPI candidates, we also collected 2570 essential human genes from the Database of Essential Genes (version 6.5)[59]. Here an essential PPI was defined as both genes of the PPI candidate being essential. In addition, we qualified our predicted PPIs and compared performances between our methods, the STRING database, the FpClass method, and the generalized interologs mapping method, based on these sets (Supplementary Note 1).

**KEGG pathway set.** To evaluate how MPs would be involved in certain kinds of pathways during tumorigenesis, we first collected 292 human pathways containing 8962 proteins from the KEGG database and then derived 22 cancer pathways belonging to the categories "Cancers: Overview" (e.g., viral carcinogenesis) and "Cancers: Specific types" (e.g., colorectal cancer-related pathway) in human diseases. Next, some pathways that were linked to these cancer pathways were regarded as related pathways. For example, there were 30 related pathways, such as the cell cycle, apoptosis, and adherens junction, recorded in the pathways in cancer (hsa05200). Finally, these 22 cancer pathways and 43 non-redundant related pathways were deemed cancer-related pathways (total of 65; Supplementary Table 1).

**Gene expression data sets in 15 cancer types.** To evaluate co-expression enrichment between genes of MPs (or MPP communities) and cancer-related pathways in 15 cancer types, we first identified DEGs between tumor tissues and corresponding normal tissues in distinct cancers. RNA-seq profiling data, including 5922 tumor samples and 660 normal tissues in 15 cancers (Supplementary Table 2), were assembled from TCGA Data Coordinating Center using the ProcessRNASeqData function of TCGA-assembler[60]. We downloaded level 3 RNA-SeqV2 data containing the expression profiles of 20,531 genes with Entrez Gene IDs for 6582 samples, and the values represented upper quartile-normalized RNA-seq by expectation maximization count estimates. Next, the counts were $\log_2$-transformed before being used for further analysis. RNA-seq data were matched through the patient barcode provided by TCGA. In addition, we assembled microarray expression data sets in these 15 cancers from GEO[20] as independent sets (Supplementary Table 5) to validate concordance for the enrichment of co-expression between the microarray and TCGA RNA-seq sets. For microarray data, the SOFT format file and corresponding annotation file retrieved from GEO were used to determine and describe the array platform, including the Probe ID, Entrez Gene ID, UniProt accession numbers, and gene description.

The following normalization strategy for gene expression sets derived from diverse microarray platforms was applied to unify the data. For Affymetrix data, we downloaded and normalized raw CEL files with the Robust Multi-array Average algorithm[61,62] (affy package v. 1.46.1 of Bioconductor v. 3.1 in R 3.2.1). Regarding probe set summarization, a custom chip definition file was used to map array oligonucleotides to the Entrez Gene ID. For Agilent data, raw TXT files were downloaded and processed with the limma package (v. 3.24.15)[63]. Background correction was performed with the backgroundCorrect function using the normexp method and an offset of 50, and normalization was implemented with the normalizeBetweenArrays function with the quantile method. Finally, a modified $t$-statistic (limma package v. 3.24.15) was utilized to measure DEGs between tumors and corresponding normal samples in each cancer type for the microarray and TCGA RNA-seq sets. The adjusted $P$ value was used for multiple hypothesis testing using Benjamini and Hochberg's method[64], and the false discovery rate was controlled at 5%. Here we used $|\log_2(\text{fold change})| \geq 1$ and adjusted $P$ values ≤0.05

to identify DEGs. The expression profiles of the DEGs were used to calculate enrichment $P$ values between MPP communities and their regulated pathways.

**A systematically integrated method for predicting MP PPIs.** To infer protein candidates interacting with an MP, we developed a SIM (Fig. 1a and Supplementary Fig. 2a–d) to calculate their interaction scores ($S_{SIM}$). $S_{SIM}$ consists of the interacting region similarity ($S_{irs}$), the quality of the PPI template ($S_{qul}$), the normalized joint sequence similarity ($S_{jss}$)[13,14], the normalized ranking of joint sequence similarity ($S_{rank}$)[14], the evolutionary conserved score across multiple species ($S_{es}$)[14], and the network topology score in a human PPI network ($S_{topo}$). $S_{SIM}$ is defined as

$$S_{SIM} = S_{irs} + S_{qul} + S_{jss} + S_{rank} + S_{es} + S_{topo} \qquad (1)$$

Based on our previous works[13,14,65–68], we statistically analyzed and simplified these six scores ranging from 0 to 1 (the total score $S_{SIM}$ ranges from 0 to 6). The detailed scoring method and scheme for identifying MP-interacting protein candidates are as follows. In the first stage, each sequence in the cytoplasmic region of an MP derived from the "Cytoplasmic" annotation is individually used to search PPI templates (i.e., reported PPIs or direct physical PPIs) by using the $S_{irs}$ (Supplementary Fig. 2a–c). The $S_{irs}$ is given as

$$S_{irs} = \sqrt{SI \times \frac{L}{Q_d}} \qquad (2)$$

where SI is the BLASTP sequence identity between sequences of the cytoplasmic region $d$ of an MP and the protein $p$ of a PPI template, $Q_d$ is the sequence length of $d$, and $L$ is the aligned length between the sequences of $d$ and $p$. Next, $S_{qul}$ is used to determine the quality of a PPI template based on the numbers of interaction detection methods ($x_m$), interacting types ($x_t$), and references ($x_r$). The $x_m$ and $x_t$ values are derived from public PPI databases and recorded using PSI MI 2.5 ontology[55]. The $x_r$ value is calculated by the number of PubMed identifiers recorded in public PPI databases. The $x_j$ value is given as

$$x_j = \begin{cases} x_j, & x_j < 2 \\ 2, & x_j \geq 2 \end{cases} \qquad (3)$$

where j is m, t, or r. Then the $S_{qul}$ is defined as

$$S_{qul} = \frac{x_m + x_t + x_r}{6} \qquad (4)$$

where $x_m$, $x_t$, and $x_r$ range from 0 to 2, and the value of $S_{qul}$ ranges from 0 to 1. In the second stage, for the selected PPI template (A–B) using $S_{irs}$, we first identified its homologous PPI candidates (i.e., A′–B′, one of them is the MP) by considering the homologous proteins (BLASTP $E$ value ≤$10^{-10}$) of proteins A and B with joint sequence similarities (joint $E$ value ≤$10^{-40}$), defined as the geometric mean of individual $E$ values of a protein pair[13,17], by searching the UniProt complete human proteome database (Supplementary Fig. 2a). According to our previous studies[13,65], the concept of homologous PPIs is briefly described as follows: (1) proteins A′ and B′ are the homologs of A and B, respectively; (2) the protein pairs A′–B′ and A–B share significant interface similarity. Then we further evaluated the $S_{jss}$ between the PPI candidate (A′–B′) and PPI template (A–B). $S_{jss}$ is given as

$$S_{jss} = \sqrt{\frac{-\log_{10}(E_{A'})}{-\log_{10}(E_A)}} \times \sqrt{\frac{-\log_{10}(E_{B'})}{-\log_{10}(E_B)}} \qquad (5)$$

where $E_{A'}$ is the BLAST $E$ value between A and A′, $E_{B'}$ is the BLAST $E$ value between B and B′, and $E_A$ and $E_B$ are the BLAST $E$ values when aligning A to A and B to B, respectively. We used $E_A$ and $E_B$ as the maximum values to normalize the joint sequence similarity ($0 \leq S_{jss} \leq 1$) because the maximum BLAST $E$ value is dependent on the protein length. $S_{rank}$ is calculated as

$$S_{rank} = 1 - \frac{\log_{10}(r_{A'-B'})}{\log_{10}(r_{max})} \qquad (6)$$

where $r_{A'-B'}$ is the rank of candidate A′–B′ based on $S_{jss}$, and $r_{max}$ is the total number of PPI candidates derived from the PPI template A–B. $S_{es}$ is defined as

$$S_{es} = \sum_{f=1}^{n} \left( E_{A'-B'}^f \times \frac{m_f}{2} \right), m_f = \begin{cases} m_f, & m_f < 2 \\ 2, & m_f \geq 2 \end{cases} \qquad (7)$$

where $E_{A'-B'}^f$ is the normalized evolutionary distance (Supplementary Fig. 2d) between the target organism (e.g., *Homo sapiens*) and the source organism $f$ (e.g., *Caenorhabditis elegans*), $n$ is the number of source organisms containing at least one PPI template used to infer the PPI candidate A′–B′, and $m_f$ is the number of PPI templates inferring the candidate A′–B′ in the source organism $f$. In this study, we assumed that the candidate A′–B′ derived from multiple PPI templates (i.e., $m_f \geq 2$) in the source organism was more highly evolutionarily conserved than that derived from only one PPI template (i.e., $m_f = 1$). This distance was obtained based on the phylogenetic tree with 273 species proposed by InParanoid[69]. In addition, the distance is the mean distance between a target organism and two corresponding source organisms if two proteins of a PPI template belong to different organisms.

Next, we computed $S_{topo}$ based on the assumption that two proteins with more shared interacting proteins, one of which has a high degree (e.g., hubs) in the network, would be more likely to associate with each other. $S_{topo}$ is calculated as

$$S_{topo} = \sqrt{\frac{C}{N} \times D} \qquad (8)$$

where $C$ is the number of shared interacting proteins between A′ (an MP) and its interacting protein B′ in the PPI network of target species (here it is *H. sapiens*); $N$ is the degree of A′ in the PPI network; $D$ is given as $1 - ((R_B - 1)/R_{max})$, which is the normalized ranking score for the fractional ranking ($R_B$) of candidate B′ degree in the PPI network based on the Borda count strategy[70], and $R_{max}$ (here it is 13,913) is the fractional ranking of the protein with the smallest degree in the network. The ranking score is normalized to avoid the long-tail phenomenon caused by certain proteins, such as polyubiquitin-C (UBC), being associated with a large number of interacting proteins that is far greater than the remaining proteins in the PPI network derived from reported PPI data.

Finally, we utilized the precision, recall, and $F_2$ measures to determine the threshold of $S_{SIM}$ for inferring MP-interacting proteins by using the SP and SN sets. Here precision and recall are defined as TP/(TP + FP) and TP/(TP + FN), where TP, FP, and FN are the numbers of true-positive, false-positive, and false-negative cases, respectively. Systematic parameter variation provided evidences that the reported PPI- and direct PPI-based $S_{SIM}$ thresholds were set to 3.6 and 3.7, respectively (Fig. 2a, Supplementary Fig. 4, and Supplementary Table 4).

**Cancer membrane protein-regulated networks.** For construction of the CaMPNets of each MP with its interacting proteins (MPP community) in 15 cancer types, we evaluated the enrichment of co-expressed gene pairs between all genes of MPP communities and cancer-related pathways using the gene expression profiles of tumor samples in TCGA RNA-seq or microarray data. For each cancer type, two DEGs with Pearson's correlation coefficient |Pearson's $r$| ≥ $h$ (here, $h$ is set to 0.5 based on a large effect size) were considered a co-expressed pair. For each DEG as an involved gene in the MPP community, we first used the co-expressed pairs between it and a cancer-related pathway to determine its involvement ($-\log_{10}$ enrichment $P$ value) for this pathway in each cancer type based on hypergeometric distribution. Moreover, for each MPP community, we measured the involvement between its involved genes and all the DEGs of regulated pathways in a certain cancer type. Here we computed the enrichment $P$ value of the hypergeometric distribution[67,71] as

$$P = \sum_{i=x}^{n} \frac{\binom{M}{i}\binom{N-M}{n-i}}{\binom{N}{n}} \qquad (9)$$

where $i$ and $n$ are the numbers of co-expressed gene pairs and all the combinational gene pairs, respectively; $x$ is the observed co-expressed gene pairs with |Pearson's $r$| ≥ 0.5, for example, $x$ and $n$ are separately 6 (orange lines) and 44 between the CHRNA9 community (two involved genes: EGFR and ERBB2) and the adherens junction pathway (comprising 22 DEGs) in BRCA (Supplementary Fig. 2e); $M$ and $N$ are the total numbers of all the co-expressed gene pairs and combinational gene pairs, respectively, between all the involved DEGs of the MPP community and all the DEGs in 292 KEGG pathways.

To further examine the statistical significance of the involvement of an MPP community regulating a specific pathway (called the observed MPP community-regulated pathway), we generated its 1000 shuffled MPP communities (Monte Carlo trials) by randomly shuffling its interacting proteins with 2594 MPs and then calculated their involvement values (an example in Supplementary Fig. 14a) for each cancer. Based on these 1000 shuffled MPP communities, we then determined the empirical $P$ value of the involvement of this MPP community-regulated pathway for a cancer. Finally, the involvement of the observed MPP community-regulated pathway in each tumor type was considered statistically significant when its empirical $P$ value was ≤0.05.

To assess each MPP community for a certain pathway with involvement significance across 15 cancer types, the enrichment $P$ value for each MPP community-regulated pathway of CaMPNets was transformed to z-score, and then these z-scores in 15 cancers were further summarized using Stouffer's unweighted Z-transform test[72] (i.e., meta-z-score; Supplementary Fig. 17a). Furthermore, we wanted to observe cancer-wide common signatures based on two issues: which cancer-related pathway is regulated by the most communities in multiple tumors, and which MPP community is involved in the most cancer-related pathways across human cancers. Therefore, we combined the meta-z-scores into a global meta-z-score for certain MPP communities (or pathways) considering 65 cancer-related pathways (or 1862 MPP communities) based on Stouffer's method (Supplementary Fig. 17c, d).

**Quantification of tumor homogeneity in CaMPNets.** To analyze the significance of the fraction of MPP community-regulated pathways shared by distinct tumor types, we first randomly shuffled gene labels of all the proteins interacting with 2594 MPs for each MPP community to generate 1000 shuffled MPP communities for each cancer-related pathway in a cancer type (again, using empirical $P$ value

≤0.05; Supplementary Fig. 14b). For each cancer-related pathway, we then calculated the fraction of MPP communities (an observed set) and the median fraction of shuffled MPP communities (expected sets) that regulate this pathway and are shared by at least 2, 3, 5, 7, and 9 cancer types (Fig. 2b, c and Supplementary Fig. 15). Finally, we used the Wilcoxon signed-rank test to compute the $P$ value between the fraction distributions of MPP communities and shuffled ones for 65 cancer-related pathways filtered at enrichment $P$ value ≤ 0.05, 0.01, 0.005, 0.001, 0.0005, or 0.0001.

**Prognostic genes and gene sets in 15 cancers**. To assess the association of each gene/gene set with survival outcomes, we only considered the patient samples (i.e., primary solid tumors) with the gene expression data and clinical outcome data (Supplementary Table 2). For each cancer, we assessed the association of each involved gene and gene set of an MPP community (or community-regulated pathway) with 10-year survival outcomes by Cox proportional hazards regression analysis using the coxph function of the R survival package (v. 2.37.2). Here we defined the involved gene set in a community-regulated pathway, containing all genes of co-expressed DEG pairs between the MPP community and the regulated pathway, for each cancer. Cox coefficients, $P$ values (log-rank test), $z$-scores, and hazard ratios (HRs) with 95% confidence intervals were acquired for each gene. We integrated all the genes of an involved gene set into a combined score ($CS_t$) in a certain cancer type with independent weights via considering their expression values in tumor samples and HRs. Here the $CS_t$ for an involved gene set with $g$ genes is defined as

$$MV_t = \sum_{j=1}^{g} \left( w_j \times E_j \right) \quad (10)$$

$$CS_t = MV_t \times RC, \quad RC = \begin{cases} -1, & \geq 75\% \text{ of the patients with } MV < 0 \\ 1, & \text{others} \end{cases} \quad (11)$$

where $E_j$ is the expression value of gene $j$ in tumor sample of the patient $t$, and $w_j$ is 1 and −1 when gene $j$ has HR ≥ 1 and <1, respectively. The weight ($w_j$) is set to 1 or −1 according to the HR of each gene to evade neutralization between adverse and favorable prognostic genes. To prevent most genes with favorable prognostic associations resulting in a large negative value and being misjudged as a low expression value by the coxph function, we set the reverse coefficient (RC) to −1 when ≥75% of the patient tumor samples had MV values <0. Note that gene sets containing 26–74% of patient tumor samples with MV ≥ 0 or <0 were considered unable to evaluate adverse or favorable prognostic associations.

For an involved gene set of the MPP community or community-regulated pathway in a cancer, the median value (50%) or auto-select best cutoff (25–75%) value of the CS was used to stratify corresponding patients into high- and low-risk groups for subjection to Kaplan–Meier analysis of their association with 10-year survival. Moreover, Cox proportional hazards regression analysis was also utilized to obtain the Cox coefficients, HRs with 95% confidence intervals, $P$ values, and $z$-scores for each gene set in a malignancy. Similarly, we assessed the cancer-wide prognostic significance for each gene set by summarizing the $z$-scores in 15 cancers into meta-$z$-scores using Stouffer's method (unweighted). We further examined cancer-wide prognostic signatures for certain MPP communities and pathways across 65 cancer-related pathways and 1862 MPP communities, respectively (Fig. 4e, f and Supplementary Fig. 26). Here we used a global meta-$z$-score to combine the meta-$z$-scores of prognostic significance using Stouffer's unweighted $Z$-transform method.

**Drug repurposing for discovering CHRNA9 inhibitors**. To discover potential CHRNA9 drugs, we applied our previous concepts and tools (e.g., Homopharma[44] and iGEMDOCK[45]) to screen 1543 FDA-approved drugs on 33 protein–ligand nAChR structures. A homopharma of protein–ligand complex comprises a set of proteins that possess a conserved sub-binding environment at protein–compound interfaces and a set of compounds with similar topology (Supplementary Fig. 34a). For the homopharma of acetylcholine-binding complex (e.g., PDB code: 1UW6[73] [http://sci-hub.tw/10.2210/pdb1UW6/pdb]), we first rapidly search for its similar binding interfaces, which consists of a set of spatially discontinuous pharma-motifs, using 3D-BLAST[74]. Here a pharma-motif is defined as a short conserved peptide forming a specific interface that has specific physico-chemical properties. Next, we superimposed these candidates to the target protein using DALI[75], a protein structure alignment tool, based on these discontinuous pharma-motifs, and retained structures with root square deviations ≤3 Å. In addition, similar compounds were also superimposed into the ligand in the target complex. Finally, we mined conserved binding environments forming conserved contact residues and similar functional groups between proteins and compounds. We next used our in-house tool iGEMDOCK to dock each drug in the FDA library to the target structures (e.g., 1UW6) for drug repurposing. The 40 top-ranked compounds with low energy were used to compute a protein–drug interaction profile based on different interaction energy types (electrostatic, hydrogen bonding, and van der Waals). According to this profile, these compounds were clustered into five groups by two-way hierarchical clustering (e.g., nicotine and bupropion were in groups 3 and 5, respectively; Supplementary Fig. 34b, c). Finally, for each group, we selected representative drugs having low binding energy and fitting the conserved binding environment.

**Cell culture and patient samples**. All human breast cancer samples were obtained from anonymous donors at Taipei Medical University Hospital, Taipei according to a protocol approved by the Institutional Review Board (N201612082). Upon histological inspection, all patient samples consisted of >80% tumor tissue. Human mammary gland epithelial cancer cells of HER2-enriched (SKBR3, AU565, BT474, UACC893, HCC1954, and HCC1419) and TNBC (Hs578T, MDA-MB-231, BT549, HCC1937, MDA-MB-436, and MDA-MB-468) cancer cell lines, a human lung cancer cell line (A549), a human hepatocellular carcinoma cell line (Hep3B), and a human normal mammary gland epithelial fibrocystic cell line (MCF-10A) were purchased from American Type Culture Collection (ATCC, Manassas, VA, USA). The human urinary bladder cancer cell line (RT4) and human pancreas cancer cell line (MIA PaCa-2) were purchased from the Bioresource Collection and Research Center (BCRC, Hsinchu, Taiwan). All cancer cells used in this study were maintained in Dulbecco's Modified Eagle's Medium (DMEM)/F12 culture medium, whereas MCF-10A cells were maintained in complete MCF-10A culture medium, which comprised a 1:1 mixture of DMEM and Ham's F12 medium supplemented with 10 μg ml⁻¹ insulin, 0.5 μg ml⁻¹ hydrocortisol, and 20 ng ml⁻¹ epidermal growth factor (Life Technologies, Rockville, MD, USA). The cell lines were confirmed to be Mycoplasma-free using Q-PCR analysis. The primer sequences are listed in Supplementary Table 8.

**Protein extraction, western blotting, and antibodies**. For the determination of protein expression, normal breast epithelial cells and other breast cancer cells were collected and listed according to breast cancer subtype. To investigate signal transduction, BT474 cells were used with 10 μM nicotine treatment for the indicated time points, whereas pretreatment with bupropion was administered 30 min before nicotine exposure. The cells were placed on ice in protein lysis buffer (50 mM Tris-HCl (pH 8.0), 120 mM NaCl₂, 0.5% Nonidet P-40 (NP-40), 100 mM sodium fluoride, and 200 mM sodium orthovanadate) containing protease and phosphatase inhibitors. Protein (50 μg) from each sample was resolved by 12% sodium dodecyl sulfate–polyacrylamide gel electrophoresis (SDS-PAGE), transferred to a nitrocellulose membrane, and analyzed by western blotting. The primary and secondary antibodies utilized are listed in Supplementary Table 7. The assay was repeated twice with duplicate samples.

**Co-immunoprecipitation (Co-IP)**. Co-IP of CHRNA9 and the associated proteins was performed on five cancer cells, including breast (BT474 and MDA-MB-231), lung (A549), bladder (RT4), and pancreatic (MIA PaCa-2) cancer cell lines. Cells were harvested in 1% NP40 lysis buffer, and 200/400 μg protein extracts were incubated with the indicated amounts of primary antibodies and Protein G beads for 3 h to allow complex formation (Supplementary Table 7), whereas IgG antibody was used as a negative control. The complexes were washed with PBS five times, denatured, and identified using SDS-PAGE immunoblotting. To avoid the presence of heavy chains from the immunoprecipitation antibody, a secondary antibody of mouse anti-rabbit light chain-specific antibody was used for CHRNA9 (or 18 interacting candidates) immunoblotting. Following protein lysis of five cancer cell lines, we immunoprecipitated protein extracts with the CHRNA9 antibody, followed by western blotting using 18 interacting partner antibodies, whereas IgG antibody immunoprecipitation served as a negative control (Supplementary Fig. 30a). In a reciprocal fashion[76], we further immunoprecipitated protein extracts with 18 interacting candidate antibodies, followed by western blotting using the CHRNA9 antibody (Supplementary Fig. 30b). To determine the positive interactions between CHRNA9 and 18 interacting candidates (or SLC16A7 and four interacting candidates), we first measured the band Intensities[77] of Immunoprecipitated Proteins (IIP) and Input loading Controls (IIC; Supplementary Fig. 30c) on blots using the ImageJ software[78]. Next, the IP ratio (IPR) for each immunoprecipitated protein in comparison to its loading control was utilized to evaluate the IP efficiency. The IPR is defined as

$$IPR = \frac{IIP}{IIC} \times \frac{AIC}{AIP} \times 100\% \quad (12)$$

where AIP and AIC are the protein Amounts used for Immunoprecipitation and Input loading Control, respectively. Here a candidate that passes the IPR threshold (>3%) of both reciprocal IP assays is considered as a positive interaction. For example, ABCB1 is a positive case in MDA-MB-231 cells (3.05 and 25.06%) but a negative case in A549 cells (2.63 and 1.61%; Supplementary Fig. 30c). To investigate formation of the CHRNA9/ERBB2 complex, cancer cells were starved for 24 h. The starved cells were then administered with nicotine for 15 min or/and administered bupropion for 30 min prior to nicotine treatment. For CHRNA9/ERBB2 complex formation, total ERBB2 was used as the loading control. All antibodies utilized for IP are listed in Supplementary Table 7.

**FRET and FLIM**. Images from FRET or FLIM experiments were performed on a Leica TCS SP5 Confocal Spectral Microscope Imaging System (Leica Microsystems, Wetzlar, Germany). For the FRET analysis of CHRNA9 and 18 candidate proteins, BT474 and MDA-MB-231 cells were hybridized with a 100× diluted primary antibodies of CHRNA9 and interacting proteins for 2 h at room temperature, followed by a 50× diluted secondary rhodamine and fluorescein isothiocyanate (FITC) dyes labeling for 1 h at room temperature, respectively. For the measurement of FRET background, BT474 and MDA-MB-231 cells were hybridized with

the primary antibody of each interacting protein (without CHRNA9 antibody), followed by secondary rhodamine and FITC dye conjugations. According to SIM strategy, CAV1 (caveolin-1) was not an interacting protein of CHRNA9; therefore, this protein pair was selected to measure the FRET efficiency as a negative control. Coverslips were mounted with VECTASHIELD Antifade Mounting Medium (Vector Laboratories, California) and imaged by confocal microscopy. The protein expression and merged images were obtained by 405- and 532-nm laser lines to excite fluorescent dyes. Then we photobleached the field at 532 nm by high-intensity light (100%) for 60 s and acquired a second set of images. The FRET efficiency (FE) was calculated as follows: FE = $(D_{post} - D_{pre})/D_{post}$ for all $D_{post} > D_{pre}$, where $D_{pre}$ and $D_{post}$ represent the donor fluorescence intensity before and after photobleaching, respectively[42,79]. At least three different cell membrane regions were examined for the presence of FRET signals. The threshold of positive interaction is defined as

$$\text{FRET}_{cut} = \max_{1 \le i \le 38} (\text{MFE}_i + \text{SE}_i) \qquad (13)$$

where $\text{MFE}_i$ and $\text{SE}_i$ are the mean and the standard error of FRET efficiency for the background signals or the negative control signal (i.e., CAV1), respectively; and $i$ is one of the 18 interacting candidates (e.g., ERBB4) and the negative control in BT474 or MDA-MB-231 cells (Supplementary Fig. 31c–f). Here the $FRET_{cut}$ was set to 0.045 (i.e., ERBB4 in MDA-MB-231). For the live cell-based FLIM, MDA-MB-231 breast cancer cells were co-transfected with the ERBB2-YFP/CHRNA9-CFP plasmids using electroporation and seeded in the glass bottom of 3.5 cm dish. After 8 h, the cells were washed with PBS and placed with starvation medium for 24 h. Leica two-photon excitation microscopy was used and the intensity input was regulated with an amplitude modulator linked to the software system. During the experiment, the starved cells were treated with 10 µM nicotine for 30 min and washed with PBS for another 30 min, whereas the intensity images from the CFP and YFP channels were recorded every 10 min. All plasmids used in the FRET assay were Sanger-sequenced and listed in Supplementary Data 8.

**Split luciferase complementation assay.** Split luciferase vectors were constructed by inserting luciferase gene fragments of N- (Nluc) and C-terminal (Cluc) amino acid residues from 1 to 398 and from 394 to 550, respectively, and a flexible linker region into pcDNA3 (Life Technologies). The CHRNA9 and ERBB2 genes were cloned into the Nluc and Cluc-pcDNA3 plasmids, respectively. Next, $5 \times 10^6$ MDA-MB-231 breast cancer cells were co-transfected using 10 µg of the CHRNA9-Nluc/ERBB2-Cluc or CHRNA9-Cluc/ERBB2-Nluc plasmid pairs. For cellular investigation, MDA-MB-231 cells were co-transfected with the CHRNA9-Cluc/ERBB2-Nluc plasmids and seeded in a six-well dish. Luciferase activity with or without 15 min of 10 µM nicotine treatment were measured by an non-invasion IVIS. For animal study, 6-week-old female BALB/c nude (CAnN.Cg-$Foxn1^{nu}$/Crl) mice were purchased from the National Science Council Animal Center of Taipei. The animal study protocol was approved by the Laboratory Animal Center (IACUC-15-327) in the National Defense Medical Center (NDMC, Taipei, Taiwan) and the Laboratory Animal Center (LAC-201-0177) of Taipei Medical University (Taipei, Taiwan). Mice were monitored every day for food and water supply, and the health status of the animals was monitored once daily by a qualified veterinarian. This study was carried out in strict accordance with the recommendations in the *Guide for the Care and Use of Laboratory Animals* from the National Institutes of Health. The NMDC SPF is accredited by the American Association for the Accreditation of Laboratory Animal Care (AAALAC). MDA-MB-231 breast cancer cells co-transfected with split luciferase plasmids were subcutaneously injected into SCID mice ($5 \times 10^6$ cells/mouse), and IVIS-imaged after 3 days. The mice were administered 100 µg kg$^{-1}$ nicotine orally for the split luciferase assay in an animal model. All plasmids used in the split luciferase complementation assay were Sanger-sequenced and listed in Supplementary Data 8.

**Orthotropic xenograft model.** SCID mice (NOD.CB17/Icr-$Prkdc^{scid}$/NcrCrl, female, 4 weeks old) purchased from the National Science Council Animal Center of Taipei were injected subcutaneously with MDA-MB-231 ($5 \times 10^6$) cells. The mice were anesthetized with 2% isoflurane, and the mammary pads of each mouse were implanted with $5 \times 10^6$ luciferase-expressing MDA-MB-231 cells. During the experiment, all mice were randomized into six groups ($n = 5$ per group) and intraperitoneally (i.p.) injected with PBS, 100 or 200 µg kg$^{-1}$ bupropion three times per week, and with or without nicotine treatment (10 µg ml$^{-1}$) via their drinking water. The mice underwent bioluminescent imaging every week by the IVIS camera system, and the images were integrated, digitized, and displayed. Regions of interest from the displayed images were identified and quantified as total photon counts or photons using the Living Image software 4.0 (Caliper, Alameda, CA). The xenografts were weighed and either snap-frozen on dry ice and stored at −80 °C for RNA and protein analysis.

**Bioluminescent imaging.** Bioluminescent imaging was performed with a highly sensitive, cooled charge-coupled device camera mounted in a light-tight specimen box (In Vivo Imaging System—IVIS™; Xenogen)[46]. Imaging and quantification of signals were controlled by the acquisition and analysis Living Image software (Xenogen). For in vivo imaging, animals were administered the substrate D-luciferin by i.p. injection at 100 mg kg$^{-1}$ in PBS and then anesthetized (2.5%

isoflurane). Mice were then placed onto a warmed stage (37 °C) inside the light-tight camera box with continuous exposure to 1.5% isoflurane. Imaging times ranged from 3 to 8 min depending on the split luciferase plasmid combination being analyzed. Generally, one group of mice (5 mice in each group) were imaged at a time. No data was excluded from the analysis. The low levels of light emitted from bioluminescent tumors or cells were detected by the IVIS camera system, integrated, digitized, and displayed. Regions of interest from the displayed images were identified around the tumor sites and quantified as photon counts per second using the Living Image software (Xenogen).

**mRNA microarray assay and Q-PCR.** To investigate our CaMPNets for studying the mechanisms of anti-metastasis on bupropion, we performed microarray analysis for mammary tumors of xenograft mice. Total RNA was extracted from the xenograft tumors using TRIzol Reagent through chloroform/isopropanol purification. RNA was labeled with 5 µg of Cy5-labeled aminoallyl RNA and then hybridized in duplicate to the Human OneArray ver. 7 release 1.0 (HOA7.1; Phalanx Biotech Group, Hsinchu, Taiwan), containing 28,264 probes with each probe corresponding to the annotated genes and proteins in the RefSeq v70 and UniProt databases, respectively. Each probe was a 60-mer oligonucleotide designed in the sense direction. Raw data were normalized using the normalizeQuantiles function of the limma package (v. 3.30.7) and log$_2$-transformed. DEGs, including upregulated and downregulated genes, were defined as those with a fold change of at least 1.5 compared with the control and were used to analyze pathway enrichment calculated by hypergeometric distribution (Supplementary Figs. 28b, c, 29, and 40–42). For Q-PCR analysis, the specific primers for each gene were synthesized (Supplementary Table 8) with the LightCycler thermocycler (Roche Molecular Biochemicals, Mannheim, Germany). All mRNA fluorescence intensities were measured and normalized to β-glucuronidase (GUS) expression using built-in software (Roche LightCycler Version 4).

**CRISPR/Cas9 gene editing of CHRNA9, ERBB2, and SLC16A7.** Custom sgRNAs for CHRNA9, ERBB2, and SLC16A7 were designed using the MIT CRISPR Design website (http://crispr.mit.edu). Guide oligonucleotides were phosphorylated, annealed, and cloned into the BsmBI site of the lentiCRISPR v2 vector (Addgene, 52961, kindly provided by Feng Zhang) according to the Zhang laboratory protocol[80] (F. Zhang lab, MIT, Cambridge, MA, USA). All plasmid constructs were verified by sequencing. Lentiviral particles were produced by transient transfection of Phoenix-ECO cells (CRL-3214) using TransIT-LT1 Reagent (Mirus Bio LLC, Madison, WI, USA). The lentiCRISPR construct was co-transfected with pMD2.G (Addgene plasmid #12259) and psPAX2 (Addgene plasmid #12260, both kindly provided by Didier Trono, EPFL, Lausanne, Switzerland). Lentiviral particles were collected at 36 and 72 h and then concentrated with a Lenti-X Concentrator (Clontech, Mountain View, CA, USA). The lentivirus concentration for each gene was quantified by Q-PCR. Biohazards and restricted materials were used in this study in accordance with the Safety Guidelines for Biosafety Level 1 to Level 3 Laboratory. The protocol was approved by the Institutional Biosafety Committee of Taipei Medical University, Taipei, Taiwan. The CRISPR/Cas9 gene-editing sequences for CHRNA9, ERBB2, and SLC16A7 are listed in Supplementary Table 8.

**Measurement of HBsAg.** Hep3B cells and CRISPR knockdown cells were seeded with culture medium in 24-well plates at a concentration of 50,000 cells per well. After the cells were attached, the culture medium was replaced with serum-free medium for 48 h, and cell numbers were determined by trypan blue exclusion. The HBsAg titer in the serum-free medium were determined by enzyme-linked immunosorbent assay (General Biological, Taiwan, Republic of China). The optical density values were normalized to the cell numbers.

**Reporting summary.** Further information on research design is available in the Nature Research Reporting Summary linked to this article.

## Data availability

The data generated in this study are available on the website of CaMPNets (http://campnets.life.nctu.edu.tw). The accession number for microarray data reported in this paper is GEO: GSE105445. The source data underlying Figs. 2, 3, 4a–c, e, f, 5a, 6d, and 7b–g and Supplementary Figs. 30a, 31f, and 32a are provided as a Source Data file. Other Supplementary Figures and other images of this study are available from the corresponding authors upon reasonable request.

## Code availability

The custom codes of SIM and CaMPNets are available on the website of CaMPNets (http://campnets.life.nctu.edu.tw).

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

## Acknowledgements

We are thankful for discussions with Ming-Daw Tsai and Andrew H.-J. Wang in Institute of Biological Chemistry, Academia Sinica (Taiwan). This work was supported by the MOST Joint Research Center for AI Technology and All Vista Healthcare (MOST107-2634-F-009-012 and MOST107-2634-F-002-019 to J.-M.Y.), Ministry of Science and Technology (MOST106-2314-B-038-053-MY3 to S.-H.T.; MOST105-2320-B-038-053-MY3 and MOST106-2632-B-038-001 to Y.-S.H.), National Health Research Institutes (NHRI-EX105-10504PI to J.-M.Y.), Ministry of Health and Welfare (MOHW106-TDU-B-212-144001 and MOHW107-TDU-B-212-114014 to Y.-S.H.; MOST102-2320-B-038-039-MY3 to C.-H.L) and Center For Intelligent Drug Systems and Smart Bio-devices (IDS$^2$B) (to Y.-H.W.L. and J.-M.Y.) and TMU Research Center of Cancer Translational Medicine (to Y.-S.H.) from The Featured Areas Research Center Program within the framework of the Higher Education Sprout Project by the Ministry of Education (MOE) in Taiwan. This work was also supported by the Taiwan Protein Project (Grant No. AS-KPQ-105-TPP). C.-Y.L. also thanks Tatsuya Akutsu in Bioinformatics Center, Institute for Chemical Research, Kyoto University (Japan), and the JSPS International Research Fellowship (ID: P17353) for support.

## Author contributions

C.-Y.L. and J.-M.Y. conceived SIM and CaMPNets, designed the framework, developed strategies for implementation and optimizations in related bioinformatics experiments, and analyzed the data. C.-H.L. and Y.-S.H. conceived and conducted all cell and animal experiments and analyzed the data. C.-Y.L., J.-M.Y, C.-H.L., and Y.-S.H. wrote the manuscript. Y.-H.C. and C.-Y.L. implemented web infrastructure for hosting CaMPNets. C.-Y.L., Y.-H.C., J.-Y.L., Y.-Y.C., and S.-H.H. collected and curated the primary data. Y.-H.W.L., Y.-J.J, and J.-K.H. commented on the manuscript at all stages. L.-C.C., C.-H. W., and S.-H.T. commented on the manuscript for clinical discussion. All authors discussed the results and their implications and were involved in manuscript editing.
