## [Peer Review File · Nature Communications]

Reviewers' comments:

Reviewer #1 (Remarks to the Author):

Membrane proteins (MPs) play critical roles in a wide range of physiological processes, including serving as receptors for signal transduction in normal and cancer cells. Identifying and accurately predicting MP-mediated protein-protein interactions (PPIs) is expected to have significant impact on the broad field of biological investigations. Therefore, it is highly significant to develop a high confidence, computational approach to predict PPIs to guide biological and therapeutic studies. The confidence in such data is often associated with high levels of experimental validation. The manuscript by Lin et al reports a computational approach to predict protein-protein interactions for transmembrane proteins with a systematically integrated method to generate a resource of Cancer Membrane Protein-regulated NETWORKS termed CaMPNETs. Extensive computational data mining and analysis were carried out to generate CaMPNETs, containing 64,125 MP-associated PPIs, which predicted numerous MP-mediated pathways and implicated potential prognostic biomarkers across 15 tumor types.

- The value of this large dataset would be significantly enhanced, to the scientific community, if authors could provide rigorous experimental evidence in support of the validity of the predicted PPIs and pathways in a selective and systematic way.
- Another area for improvement is to make efforts to distinct integrated PPIs based on direct physical interactions from those indirect interactions based on functional correlations. Without such a clear definition, more datasets may result in more confusions in the field, leading to increased challenges for systems biology studies.
- Are there any MP-specific PPI network properties?
- Even though the CaMPNETs was intended to establish the importance of PPIs in cancer by the authors, most data for functional correlations were generated from expression data sets for genes in the "community". The novelty and advantage of the PPI networks for inferring functional relevance was unclear.
- After the description of computational predictions of MP binding partners, authors focused on functional validation of newly identified PPI between CHRNA9 and ERBB2 in breast cancer cells and in a mouse model. Limitations in experimental design do not allow us to conclude whether this reported PPI has any functional impact on tumor progression. For example, to determine functional relevance of the CHRNA9/ERBB2 PPI, the authors utilized nicotine and bupropion as ligands for CHRNA9. The observed effects are not necessarily due to the regulation of the interaction, but can be induced by other mechanisms regulated by CHRNA9. Furthermore, both compounds are not specific to CHRNA9 and target the whole family of nAChRs, and bupropion has other well established targets. It is interesting that nicotine and bupropion modulate the interaction between CHRNA9 and ERBB2. However, the IP assay lacked critical controls. To demonstrate the functional relevance of the CHRNA9/ERBB2 PPI, additional experiments with tools to manipulate this interaction are required.
- It is inconclusive how many of the 18 candidate partners for CHRNA9 could be validated in orthogonal assays. This is important for the estimation of the validation rate of the predicted PPIs. It is unclear how "novel PPIs" were defined. At least two "new" binding partners as described (SRC and FYN, Figure 6A) are known binding partners of nicotinic acetylcholine receptors (see JBC Vol. 280, pp. 25928–25935, 2005; JBC Vol. 279, pp. 8779–8786, 2004; J Neurosci Res. 1996 Jun 1;44:421-9.).

Additional comments:

- A brief comparison with available approaches would be helpful to readers to identify potential advantages. Also prediction accuracy for different approaches should be evaluated, discussed, and summarized in a separate table.
- The functional association between the MPs and predicted binding partners could be strengthened using the experimental data from the recent loss-of-function screens (e.g. Project Achilles data).
- It is mentioned in Methods that "The positive cases of MPs in the SP set consisted of 17,736

human PPIs derived from the experimental PPI set." Clarify what the "experimental PPI set" is.

- Authors built a set of "template" experimentally detected, physical PPIs. It is mentioned that, "To predict proteins that interact with the MPs, we first collected 749,087 non-redundant PPIs across 497 species (called the experimental PPI set) from five public databases, including IntAct57, BioGRID58, DIP59, MIPS60, and MINT61, and then filtered by considering both two proteins of each PPI recorded in the UniProt complete proteome database." What/Any settings/filters were used to extract physical direct PPIs? What is the overlap between PPIs reported for the human and other organisms? Did authors include PPIs detected with high-throughput assays (e.g. mass-spec, Y2H) or more detailed conventional assays, or both? What is the overlap between the reference databases? The Sqi score takes into account the number of methods by which a PPI was detected but not the nature of those methods. For example, it is clear that evidences from endogenous co-IP in cancer cells and high throughput mass-spec data are not equivalent.
- Authors searched the human proteome to find proteins similar to the "template" PPIs. Are "Template PPIs" "known binding partners"?
- Predictions were compared with STRING database. It is unclear how this comparison was performed. Did authors use an independent test set of KNOWN PPIs included in STRING but not included in IntAct, BioGRID, DIP, MIPS, and MINT databases that were used to train the algorithm? What is the overlap between STRING and other DBs? How did the authors evaluate the accuracy of PPIs PREDICTED by STRING?
- What is the overlap between the binding partners identified for different membrane proteins?

Reviewer #2 (Remarks to the Author):

The manuscript by Lin et al., presents a comprehensive global analysis of membrane protein interactions across multiple human tumor types using various statistical and bioinformatics analytical methods. They have mapped the interactions of membrane proteins with their partners, with an emphasis on the cytoplasmic domains of the proteins, and demonstrate how the interaction networks could predict prognosis and perhaps the occurrence of secondary tumors. Experimental proof of concept is provided for their statistical analysis methods, using nAChR as a model; they demonstrate that the interaction of $\alpha 9$ nAChR with ERBB2 is induced rapidly upon nicotine stimulation and the interaction dissipates within 3 minutes. Overall, the manuscript is interesting and the studies would be of relevance to a broad spectrum of investigators.

At the same time, there are certain issues that need to be addressed. First, it is not clear how strong the prediction of the appearance of secondary tumors is. Would it be possible to provide additional validation for their findings? Second, in a similar vein, it would significantly strengthen the manuscript if they could experimentally demonstrate some of the novel interactions across multiple tumor types, or cell lines derived from them. Third, the data on $\alpha 9$ nAChR is interesting, but there is hardly any etiological data that support a role for nicotine or smoking with breast cancer. It would be highly desirable to conduct the basic interaction studies in lung, pancreatic or bladder cancer cell lines. While the data with bupropion is interesting, it would be more relevant in a cancer that is strongly associated with smoking. Fourth, it is generally thought that it is the $\alpha 7$ subunit of nAChR that promotes tumor progression and metastasis; thus, the data with $\alpha 9$ subunit is rather surprising. Would it be possible to incorporate or discuss data obtained with this subunit? Fifth, the relevance of nAChR9 on infectious diseases and cancers of viral origin is not clear. While there is a significant amount of literature on how nAChRs affect immune response, there is no concrete evidence that nAChRs facilitate HTLV-1 infection or contribute to hepatitis. Additional support should be provided to strengthen the data in Figure 5.

Overall, studies described this manuscript are comprehensive and significant. It can be improved further by certain modifications, as suggested above.

Reviewer #3 (Remarks to the Author):

The authors describe a novel approach to uncover the roles of membrane proteins as cancer hallmarks. The study is innovative and promising and also well-drafted. However, there remain significant concerns that need to be addressed.

Major

1. The authors describe MP-community as MP community consists of an MP and its interacting proteins derived from the known and predicted PPIs. The term MP-community at the first glance it gives an impression that it consists exclusively of MPs, but actually in vivo MPs interact both with other MPs and with non-MPs; reflecting this, MP communities will most likely consist of both MP and non-MP proteins. Also, intracellular domains within the membrane-anchored proteins may translocate to cytoplasm or other cellular compartments after proteolytic processing; the translocated domains may then interact with other non-MP proteins. Can the authors clarify the makeup of the MP communities and the percentage of membrane-embedded proteins in each one of them? Is it feasible to refer to an MP-community as such if it's made up of largely non-MP proteins?

2. Secondly, the authors have extracted 749,087 non-redundant PPIs from public repositories and compiled a set of PPIs for constructing MP-communities. However, they do not appear to have distinguished binary interactions from co-complex memberships in this set. In the absence of such distinction, the authors' inferences that the inferred networks display scale-free topology and their data on the identification and properties of MP community hubs are untenable.

3. Materials and Methods: KEGG pathway set- The authors claim that 22 cancer pathways belonging to the categories 'Cancers: Overview' and 43 related pathways (e.g., cell cycle and adherens junction) were deemed cancer-related pathways. What were the criteria to select the "43 related pathways" as cancer-related pathways? This must be clarified in the text.

4. Page 6: ...cancer-specific CaMPNets... It is a little confusing to label the inferred CaMPNets as cancer-specific, unless the authors can establish that these CaMPNets are not observed in non-cancer datasets. Especially since a large number of cancer-related pathways that were included in the analysis participate in non-cancer physiological processes.

5. CaMPNets are constructed based on the enrichment of co-expressed gene pairs between DEGs and specific cancer-related pathways. Have the authors examined if the co-expressed genes lie within close proximity to each other in the correlated pathways? This is because co-expression maybe less evident in the case signalling pathways that are often hierarchical in nature and as such the author's approach may miss out on gene sets that belong to the same pathway but may not be sufficiently co-expressed.

Minor

1. What were the criteria for selecting the 15 cancer types, why were only these cancers considered for the analysis?

2. Page 4, Results "Identification of proteins interacting with MPs"- ...reflecting that many MP-interacting proteins still unknown. -> ...reflecting that many MP-interacting proteins are still unknown.

3. Page 5, presented essential properties in humans-> performed/participated in essential properties in humans

We have carefully revised the manuscript according to all the suggestions from the reviewers. Our point-to-point responses are included as below. **Please note that all changes in the revised manuscript are marked red to facilitate the review.**

Reviewer #1:

Major comments:

1. The manuscript by Lin et al reports a computational approach to predict protein-protein interactions for transmembrane proteins with a systematically integrated method to generate a resource of Cancer Membrane Protein-regulated NETWORKs termed CaMPNets. Extensive computational data mining and analysis were carried out to generate CaMPNets, containing 64,125 MP-associated PPIs, which predicted numerous MP-mediated pathways and implicated potential prognostic biomarkers across 15 tumor types.

Reply: Thank you for your comments.

2. The value of this large dataset would be significantly enhanced, to the scientific community, if authors could provide rigorous experimental evidence in support of the validity of the predicted PPIs and pathways in a selective and systematic way.

Reply:

In this revised manuscript, we provided additional, more rigorous experimental evidence in support of the validity of the predicted PPIs and pathways in a selective and systematic way.

First, we performed orthogonal assays of 18 representative interacting partners of CHRNA9 on breast (MDA-MB-231 and BT474), lung (A549), pancreatic (MIA PaCa-2), and bladder (RT4) cancer cell lines using immunoprecipitation (IP) or Förster resonance energy transfer (FRET) experiments. Second, we also illustrated that CHK1, CDK1 and PLK1 were associated with SLC16A7 in MDA-MB-231 and A549 cancer cells. Third, a clinical investigation of HER2+ breast tissues revealed a strong protein interaction between CHRNA9 and ERBB2 compared to that in TNBC breast tissues (**Fig. 6c**). (Page 17, line 11 to page 18, line 13)

Moreover, we also addressed “**Comment 9**” by using experimental data from the loss-of-function screens from the Project Achilles data¹ to investigate the functional association between the MPs and predicted binding partners (details in the reply of **Comment 9**). (Page 6, lines 2-8; details in **Supplementary Note 4**)

Additionally, we performed microarray analysis to examine the effects of CHRNA9 and SLC16A7 knockdown in MDA-MB-231 and A549 cell lines treated with CRISPR/Cas9 targeting the CHRNA9 and SLC16A7 gene loci, respectively. (Page 16, lines 21-24; details in **Supplementary Fig. 28**)

Finally, we addressed “**Comment 5**” to further clarify the novelty and advantage of the MP PPI subnetworks for inferring functional relevance between MPs and cancer-related pathways (details in the reply of **Comment 5**). (Page 8, lines 6-15; details in **Supplementary Fig. 17**)

3. Another area for improvement is to make efforts to distinct integrated PPIs based on direct physical interactions from those indirect interactions based on functional correlations. Without such a clear definition, more datasets may result in more confusions in the field, leading to increased challenges for systems biology studies.

Reply:

In this revised manuscript, we distinguished direct physical interaction data from all experimentally validated data to present both all PPI-based and direct PPI-based SIMs/CaMPNets in human cancers.

Based on the definition of a direct physical interaction in the DIP², we further used the term of direct interaction (MI:0407) and the subclass terms of direct interaction (e.g., covalent binding, MI:0195) to define an experimentally validated PPI as the direct interaction. Using this criterion, 176,087 experimentally validated PPIs belonged to the direct PPIs. (Page 40, lines 5-13)

Furthermore, using the positive sets, i.e., 18,827 MP PPIs (or 8,017 MP direct PPIs), and the negative set, we compared the PPI prediction accuracies of all PPI-based and direct PPI-based SIMs. (Page 5, lines 10-18; details in **Supplementary Note 2**)

Finally, we identified and compared all PPI- and direct PPI-based MPP community (i.e., MP partner community)-regulated pathways to obtain two global maps of the CaMPNet patterns between 1,515 overlapping MPP communities and 65 cancer-related pathways across 15 cancers. (details in **Supplementary Note 2**)

4. Are there any MP-specific PPI network properties?

Reply:

In this revised manuscript, we performed two analyses: (1) an analysis of the hub properties of MPs and other proteins (called non-MPs) in the PPI networks and (2) an analysis of the topological properties and functional enrichment of the MP-focused and non-MP subnetworks to investigate the MP-specific PPI network properties. (Page 6, lines 12-17; details in **Supplementary Note 5**)

5. Even though the CaMPNets was intended to establish the importance of PPIs in cancer by the authors, most data for functional correlations were generated from expression data sets for genes in the “community”. The novelty and advantage of the PPI networks for inferring functional relevance was unclear.

Reply:

In this revised manuscript, we redrew **Figure 3a** and added **Supplementary Figure 17** and rewrote some statements to clarify the novelty and advantage of the MP PPI subnetworks for inferring the functional relevance between MPs and cancer-related pathways.

We identified and compared the regulation between MPs and the pathways in 15 distinct cancers by considering only the MP itself, the MP with known PPIs, and the all PPI-based (or direct PPI-based) MPP community (i.e., MP partner community), including the MP and its known and predicted PPIs. (Page 8, lines 6-15; details in **Supplementary Fig. 17**)

Of course, one potential limitation of CaMPNets is that our approach may miss gene sets that belong to the same pathway but are potentially not sufficiently co-expressed, as the co-expression may be less evident in the case signaling pathways that are often hierarchical in nature (comments from Reviewer 3). Therefore, we rewrote the statements in the discussion section for describing the limitation of our approach. (Page 24, line 16 to page 25, line 3)

6. After the description of computational predictions of MP binding partners, authors focused on functional validation of newly identified PPI between CHRNA9 and ERBB2 in breast cancer cells and in a mouse model. Limitations in experimental design do not allow us to conclude whether this reported PPI has any functional impact on tumor progression. For example, to determine functional relevance of the CHRNA9/ERBB2 PPI, the authors utilized nicotine and bupropion as ligands for CHRNA9. The observed effects are not necessarily due to the regulation of the interaction, but can be induced by other mechanisms regulated by CHRNA9. Furthermore, both compounds are not specific to CHRNA9 and target the whole family of nAChRs, and bupropion has other well established targets. It is interesting that nicotine and bupropion modulate the interaction between CHRNA9 and ERBB2. However, the IP assay lacked critical controls. To demonstrate the functional relevance of the CHRNA9/ERBB2 PPI, additional experiments with tools to manipulate this interaction are required.

Reply:

In this revised manuscript, we added the **Figure 7c-f**, **Supplementary Figures 37** and **38** and **Supplementary Note 10** for discussing the functional relevance of the CHRNA9/ERBB2 PPI.

To study the regulatory mechanism of bupropion on CHRNA9 and ERBB2, we selectively knocked down CHRNA9 (CHRNA9 KD) and ERBB2 (ERBB2 KD) expression using the clustered regularly interspaced short palindromic repeats/CRISPR-associated proteins (CRISPR/Cas9) system in BT474 and MDA-MB-231 cancer cells. (Page 21, lines 3-13; details in **Supplementary Note 10**)

This study suggests that bupropion could have utility in smoking-related metastatic cancer patients with high nAChR expression. However, the target proteins (e.g., nAChRs) of bupropion

to suppress nicotine-induced complex disassociation, downstream signals, and metastatic ability in BRCA remain to be fully elucidated. (Page 24, lines 7-10)

7. It is inconclusive how many of the 18 candidate partners for CHRNA9 could be validated in orthogonal assays. This is important for the estimation of the validation rate of the predicted PPIs. It is unclear how “novel PPIs” were defined. At least two "new" binding partners as described (SRC and FYN, Figure 6A) are known binding partners of nicotinic acetylcholine receptors (see JBC Vol. 280, pp. 25928–25935, 2005; JBC Vol. 279, pp. 8779–8786, 2004; J Neurosci Res. 1996 Jun 1;44:421-9.).

Reply:

Thank you for your comments. In this revised manuscript, to experimentally validate our SIM strategy in general, we examined 18 representative interacting partners of CHRNA9 (details in **Supplementary Note 8** and **Supplementary Fig. 26a**) via immunoprecipitation (IP; **Supplementary Fig. 29**) or Förster resonance energy transfer (FRET; **Supplementary Fig. 30**) assays in human cancer cells. (Page 17, line 11 to page 18, line 3)

Additionally, we replaced "novel PPIs" by "new PPIs", which were first identified and validated in this work. APP³, EGFR⁴, FYN⁵⁻⁷, and SRC⁷⁻⁹ have been proposed to bind with other nAChRs (**Supplementary Fig. 26c**). Although there is no study indicating that 18 selected candidates interact with CHRNA9, the findings for four binding proteins of nAChRs provide their predictable associations with CHRNA9. (Page 15, lines 21-22; details in **Supplementary Fig. 26c**)

Minor comments:

8. A brief comparison with available approaches would be helpful to readers to identify potential advantages. Also prediction accuracy for different approaches should be evaluated, discussed, and summarized in a separate table.

Reply:

In this revised manuscript, we compared our SIM with the STRING database¹⁰, the FpClass method¹¹, and the generalized interolog mapping method¹² using prediction accuracies (**Fig. 2a**, **Supplementary Fig. 4**, and **Supplementary Table 4**) and their features (**Supplementary Table 3**). (Page 5, lines 6-23; Page 41, lines 1-3; details in **Supplementary Notes 1 and 2**)

9. The functional association between the MPs and predicted binding partners could be strengthened using the experimental data from the recent loss-of-function screens (e.g. Project Achilles data).

Reply:

In this revised manuscript, we added the **Supplementary Figure 8** and **Supplementary Note 4** to analyze the functional association between the MPs and their predicted binding partners using the experimental data of loss-of-function screens from the Project Achilles data. (Page 6, lines 2-8; details in **Supplementary Note 4**)

10. It is mentioned in Methods that “The positive cases of MPs in the SP set consisted of 17,736 human PPIs derived from the experimental PPI set.” Clarify what the “experimental PPI set” is.

Reply:

In this revised manuscript, we rewrote the statements for clarifying “experimental PPI set”. (Page 39, line 21 to page 40, line 17)

Please note that to evaluate the reliability of the predicted PPIs for 2,594 human MPs derived from our method, we further curated three data sets, two standard positive (SP) sets and a negative (SN) set. The positive cases of MPs in two SP sets consisted of 18,827 PPIs (here, 17,736 PPIs having proteins recorded in the human UniProt Swiss-Prot database) and 8,017 direct PPIs in humans derived from the experimentally validated PPI set. Additionally, the experimentally validated PPIs (or direct PPIs) for each MP belonged to the sets of positive cases and were not selected as the template PPIs when we evaluated and compared the predictive powers with the other methods.

11. Authors built a set of “template” experimentally detected, physical PPIs. It is mentioned that, “To predict proteins that interact with the MPs, we first collected 749,087 non-redundant PPIs across 497 species (called the experimental PPI set) from five public databases, including IntAct¹³, BioGRID¹⁴, DIP², MIPS¹⁵, and MINT¹⁶, and then filtered by considering both two proteins of each PPI recorded in the UniProt complete proteome database.

(1) What/Any settings/filters were used to extract physical direct PPIs?

(2) What is the overlap between PPIs reported for the human and other organisms?

(3) Did authors include PPIs detected with high-throughput assays (e.g. mass-spec, Y2H) or more detailed conventional assays, or both?

(4) What is the overlap between the reference databases?

(5) The Ssql score takes into account the number of methods by which a PPI was detected but not the nature of those methods. For example, it is clear that evidences from endogenous co-IP in cancer cells and high throughput mass-spec data are not equivalent.

Reply:

(1)

As state above in our reply to **Comment 3**, we used the term of direct interaction (MI:0407) and the subclass terms of direct interaction (e.g., covalent binding, MI:0195) implemented in the

PSI MI¹⁷ to define an experimentally validated PPI as the direct interaction. Based on these criteria, 176,087 experimentally validated PPIs belong to the direct interactions among a total of 749,087 PPIs. (Page 40, lines 5-13)

(2)

To examine whether the PPIs experimentally validated for humans could be observed as homologous PPIs in other organisms (i.e., overlap between PPIs reported for humans and other organisms), we assembled 44,233 homology groups across 21 organisms from the NCBI HomoloGene database¹⁸. Here, we defined a PPI A-B in humans and a PPI A'-B' in another organism as homologous PPIs in a homologous PPI group when A, A' and B, B' belonged to the respective homology groups. Among 173,327 and 55,233 homologous PPI groups containing at least one PPI and direct PPI in humans, our results showed that only 11% and 8% of these groups consisted of homologous PPI(s) in other organisms, respectively (Fig. R1-1). Moreover, we also found that 86% and 93% of the predicted PPIs were identified by the human PPI templates (i.e., the best matching templates) using the all PPI-based and direct PPI-based S_{SIM} approaches, respectively (Fig. R1-2a). Considering predicted PPIs identified by multiple templates at the given thresholds, 44% and 28% of the predicted PPIs were individually derived from more than one PPI template and direct PPI template (Fig. R1-2b). This result implies that the paralogous interactions in humans and the homologous interactions in other mammals majorly contribute to the identification of human MP PPIs, but the homologous interactions in the other organisms could also provide clues.

Figure R1-1 | Relationships between the numbers of nonhuman organisms containing homologous PPIs of the human PPIs and the numbers of homologous PPI groups.

A total of 44,233 homology groups across 21 organisms were assembled from the NCBI HomoloGene database¹⁸. Among 749,087 PPIs and 176,087 direct PPIs, 69% (515,061) of PPIs and 70% (122,784) of the direct PPIs could be assigned to one of the homologous PPI groups, respectively. Among these homologous PPI groups, 173,327 and 55,233 groups consisted of at least one PPI and direct PPI in humans. Note that no homologous PPI group contained homologous PPIs from both humans and more than seven nonhuman organisms.

Figure R1-2 | Source organism distributions of PPI templates for 130,952 and 58,862 predicted PPIs that were identified by all PPI-based and direct PPI-based SIMs, respectively.

(a) Organism distributions of the best matching PPI templates for the predicted PPIs. (b) Relationship between the number of organisms containing at least one PPI template and the percentage of predicted PPIs. In the SIM strategy, a predicted PPI could be identified by more than one PPI template from the same or different organisms, and the PPI template with the highest S_{SIM} was selected as the best matching PPI template.

(3)

Yes, 749,087 experimentally validated PPIs derived from five public PPI databases included the PPIs detected by both high-throughput assays (e.g., mass-spec, Y2H) and more detailed conventional assays. In this revised manuscript, we further distinguished direct physical interaction data from all experimentally validated data to provide a high confidence PPI data set as the direct PPI-based template set and standard positive set.

(4)

In five public PPI databases, most of the experimentally validated PPIs (97%, 728,943/749,087) were assembled from the BioGRID and IntAct databases, and only 3,422 (0.5%) PPIs were recorded in all five databases (**Supplementary Fig. 42a**). Similarly, we also observed that 99% (173,967) of 176,087 direct physical interactions were collected from the BioGRID and IntAct databases, and the overlap of all databases was 1.6% (2,824; **Supplementary Fig. 42b**). (Page 39, lines 21-23; details in **Supplementary Fig. 42**)

(5)

To examine whether the S_{qul} score, taking into account the weight of the detection method, could improve the predictive power of our SIM, we assessed the weights of detection methods according to the Human Integrated Protein-Protein Interaction rEference (HIPPIE) scores assigned to experimental techniques (HIPPIE v2.0)¹⁹. HIPPIE scores (HS s) ranging from 0 to 10 were used to weight the interaction by considering 109 detection methods. For example, coimmunoprecipitation (MI:0019) has a score of 5, and pull down (MI:0096) has a score of only 2.5. To determine the detection method confidence and reproducibility (i.e., validated in orthogonal assays) for a PPI template, we selected the top two detection methods ranked by HS and then used these methods to evaluate the x_m , which is defined as

$$x_m = \frac{\sum_{b=1}^2 (HS_b)}{10}$$

where the HS_b is set to 0 when any of the following conditions are satisfied: (1) the score of detection method b is 0 in the HIPPIE, (2) there is no detection method b , or (3) the detection method b has no HS ; and the value of x_m ranges from 0 to 2. The x_t value is derived from public PPI databases and recorded using PSI MI 2.5 ontology¹⁷. The x_r value is calculated by the number of PubMed identifiers recorded in public PPI databases. The x_j value is given as $x_j = \begin{cases} x_j, & x_j < 2 \\ 2, & x_j \geq 2 \end{cases}$

where j is t or r . Then, the weighted S_{qul} is defined as

$$S_{qul} = \frac{x_m + x_t + x_r}{6}$$

where x_m , x_t , and x_r range from 0 to 2, and the value of S_{qul} ranges from 0 to 1. Subsequently, we compared the PPI prediction accuracies of all PPI-based and direct PPI-based SIMs with or without considering the weights of the detection methods (i.e., weighted S_{qul}). Our results indicate that both S_{SIM} approaches using original S_{qul} or weighted S_{qul} yielded almost the same AUCs regardless of whether the all PPI-based and direct PPI-based SIMs and the corresponding positive sets were used (**Fig. R1-3a**).

This finding can be explained by the observation that only 2.4% of experimentally validated PPIs and 6.4% of direct PPIs had HIPPIE scores > 5 (**Fig. R1-3b**), and the HIPPIE scores for most of these PPIs (i.e., 76% of experimentally validated PPIs and 78% of direct PPIs) were five. Additionally, we found that the means of the maximum HIPPIE scores of experimentally validated

PPIs and direct PPIs were more than 5 when their detection method numbers were at least two (Fig. R1-3c). Approximately 97% of all PPIs and 98% of direct PPIs, validated by at least two detection methods (original x_j value is denoted by two), had maximum HIPPIE scores ≥ 5 . In comparison to weighted S_{qul} , the above results suggest that our original S_{qul} considering the number of detection methods could not only be used to evaluate the detection method confidence of PPIs but also exert nearly the same predictive power. Therefore, we retained the original S_{qul} as a part of our S_{SIM} .

Figure R1-3 | Performance analysis of our integrated scoring (S_{SIM}) using S_{qul} with/without considering the weights of the detection methods.

(a) ROC curves of the prediction accuracies of our S_{SIM} using S_{qul} with or without considering the weights of the detection methods (weighted S_{qul}) and all PPIs or direct PPIs as templates (T) and standard positives (P). (b) Maximum HIPPIE score distributions of 749,087 experimentally validated PPIs and 176,087 direct PPIs based on the confidence scores assigned to experimental techniques (HIPPIE v2.0)¹⁹. The HIPPIE scores of experimentally validated PPIs were 0 if their detection methods

with the maximum HIPPIE score = 0. (c) Relationship between the numbers of detection methods against the mean of the maximum HIPPIE score or the PPI numbers for all and direct PPIs.

12. Authors searched the human proteome to find proteins similar to the “template” PPIs. Are “Template PPIs” “known binding partners”?

Reply:

Yes, the template PPIs are 749,087 experimentally validated PPIs (i.e., known binding partners for 83,179 proteins) across 497 species. In this revised manuscript, we rewrote the statements for describing template PPIs. (Page 39, line 21 to page 40, line 13)

Please note that the experimentally validated PPIs (or direct physical PPIs) for each MP belonged to the set of positive cases and would not be selected as the template PPIs when we evaluated and compared the predictive powers with other methods.

13. Predictions were compared with STRING database. It is unclear how this comparison was performed.

Did authors use an independent test set of KNOWN PPIs included in STRING but not included in IntAct, BioGRID, DIP, MIPS, and MINT databases that were used to train the algorithm?

What is the overlap between STRING and other DBs?

How did the authors evaluate the accuracy of PPIs PREDICTED by STRING?

Reply:

In this revised manuscript, we added **Supplementary Figure 43** and **Supplementary Note 1** for clarifying these questions and how the comparison between our method and STRING database. (Page 5, lines 6-10; Page 41, lines 1-3; details in **Supplementary Note 1**)

14. What is the overlap between the binding partners identified for different membrane proteins?

Reply:

In this revised manuscript, we added the **Supplementary Figure 12** and **Supplementary Note 6** for analyzing the overlap between the binding partners identified for different membrane proteins. (Page 6, lines 17-21; details in **Supplementary Note 6**)

We have carefully revised the manuscript according to all the suggestions from the reviewers. Our point-to-point responses are included as below. **Please note that all changes in the revised manuscript are marked red to facilitate the review.**

Reviewer #2:

1. The manuscript by Lin et al., presents a comprehensive global analysis of membrane protein interactions across multiple human tumor types using various statistical and bioinformatics analytical methods. They have mapped the interactions of membrane proteins with their partners, with an emphasis on the cytoplasmic domains of the proteins, and demonstrate how the interaction networks could predict prognosis and perhaps the occurrence of secondary tumors. Experimental proof of concept is provided for their statistical analysis methods, using nAChR as a model; they demonstrate that the interaction of $\alpha 9$ nAChR with ERBB2 is induced rapidly upon nicotine stimulation and the interaction dissipates within 3 minutes. Overall, the manuscript is interesting and the studies would be of relevance to a broad spectrum of investigators.

Reply: Thank you for acknowledging our works.

2. At the same time, there are certain issues that need to be addressed. First, it is not clear how strong the prediction of the appearance of secondary tumors is. Would it be possible to provide additional validation for their findings?

Reply:

In this revised manuscript, we redraw the **Supplementary Figure 22** and added **Supplementary Note 7** for providing additional comparisons and validation for our findings. (Page 11, line 18 to page 12, line 2; details in **Supplementary Note 7**).

3. Second, in a similar vein, it would significantly strengthen the manuscript if they could experimentally demonstrate some of the novel interactions across multiple tumor types, or cell lines derived from them.

Reply:

In this revised manuscript, we provided additional, more rigorous experimental evidence in support of the validity of the predicted PPIs. To significantly strengthen this revised manuscript, we performed orthogonal assays of 18 representative interacting partners of CHRNA9 on breast (MDA-MB-231 and BT474), lung (A549), pancreatic (MIA PaCa-2), and bladder (RT4) cancer cell lines using immunoprecipitation (IP) or Förster resonance energy transfer (FRET) experiments. Additionally, we also illustrated that CHK1, CDK1 and PLK1 were associated with SLC16A7 in MDA-MB-231 and A549 cancer cells. (Page 17, line 11 to page 18, line 13)

4. Third, the data on $\alpha 9$ nAChR is interesting, but there is hardly any etiological data that support a role for nicotine or smoking with breast cancer. It would be highly desirable to conduct the basic interaction studies in lung, pancreatic or bladder cancer cell lines. While the data with bupropion is interesting, it would be more relevant in a cancer that is strongly associated with smoking.

Reply:

In our previous studies^{20,21}, we found that nicotinic receptor-mediated biological mechanisms play a decisive role in tumor formation. The $\alpha 9$ -nAChR (i.e., CHRNA9) expression in human breast cancer (MDA-MB-231) and normal (MCF-10A) cells was forcibly altered by RNA interference and adenovirus (Tet-Off) experiments, respectively. Our data demonstrate that the nicotine- and NNK-induced cancer cell proliferation effects were prohibited in $\alpha 9$ -nAChR-expressing (Si $\alpha 9$) cells. In contrast, the growth of MCF-10A-Nic (DOX-) tumors that expressed higher levels of $\alpha 9$ -nAChR was substantially increased compared to that of MCF-10A-Nic (DOX+) tumors that expressed normal levels of $\alpha 9$ -nAChR. Additionally, to examine the prognostic value of $\alpha 9$ -nAChR mRNA expression in breast cancer patients at different stages, we measured $\alpha 9$ -nAChR mRNA by real-time PCR in both surgical and laser capture microdissected tumor samples. Our results suggest that higher $\alpha 9$ -nAChR mRNA expression is associated with later disease stages among breast cancer patients. These results imply that receptor-mediated carcinogenic signals play a decisive role in biological functions related to human breast cancer development.

In this revised manuscript, we performed orthogonal assays of 18 representative interacting partners of CHRNA9 in breast (MDA-MB-231 and BT474), lung (A549), pancreatic (MIA PaCa-2), and bladder (RT4) cancer cell lines using immunoprecipitation (IP) or Förster resonance energy transfer (FRET) experiments (details in the reply of **Comment 3**). (Page 17, line 11 to page 18, line 13)

The inhibitory effect of the CHRNA9/ERBB2 complex disassociation and signal transduction caused by bupropion were also found in lung cancer (A549) cells exposed to nicotine. (Page 20, line 24 to page 21, line 1; details in **Supplementary Fig. 36**)

5. Fourth, it is generally thought that it is the $\alpha 7$ subunit of nAChR that promotes tumor progression and metastasis; thus, the data with $\alpha 9$ subunit is rather surprising. Would it be possible to incorporate or discuss data obtained with this subunit?

Reply:

In a previous study²⁰, we characterized the expression of nAChR subunits in normal (nonmalignant) human breast cell lines (MCF-10A and HBL-100) and human breast cancer cell lines (MDA-MB-231, MDA-MB-453, AU-565, BT-483, and MCF-7). To evaluate the expression of nAChR subunits among Taiwanese breast cancer patients, human breast tumors (n = 50) and the

surrounding normal tissues were dissected and subjected to RT-PCR separately. All breast cell lines were found to express similar ($\alpha 5$, $\alpha 9$, and $\alpha 10$) nAChR subunits (**Fig. R2-1a**). The same three nAChR subunits ($\alpha 5$, $\alpha 9$, and $\alpha 10$) predominated normal and malignant breast tissues (**Fig. R2-1b**). We found increased $\alpha 9$ -nAChR mRNA levels in nearly all tumor tissues compared with that in normal tissues (**Fig. R2-2**). In contrast, the mRNA levels of the $\alpha 5$ - and $\alpha 10$ -nAChR subunits were not substantially different between tumor and normal paired samples. We next compared the mRNA levels of the $\alpha 9$ -nAChR subunit in 276 tumor and normal paired tissue samples by real-time PCR analysis. The real-time PCR results of normal and tumor tissues were calculated, and the tumors were divided into two groups according to their $\alpha 9$ -nAChR mRNA expression patterns. Here, 186 of 276 (67.3%) normal vs tumor tissue pairs fell into the group in which $\alpha 9$ -nAChR was expressed at higher levels in tumor tissue than in normal tissue ($T > N$), and 90 paired samples had somewhat higher $\alpha 9$ -nAChR expression in normal tissue than in tumor tissue ($N > T$) (**Fig. R2-3a**). In the group with higher tumor $\alpha 9$ -nAChR expression than normal $\alpha 9$ -nAChR expression ($T > N$) overall, the $\alpha 9$ -nAChR expression in tumor cells was 7.84-fold greater than that in normal cells ($P = 0.002$). Additionally, a more than fivefold increase in $\alpha 9$ -nAChR mRNA expression was detected in 57 of 186 (30.6%) tumor tissue samples (**Fig. R2-3a** bars 3 and 4). However, in the group with higher normal $\alpha 9$ -nAChR expression than tumor $\alpha 9$ -nAChR expression ($N > T$), nearly all of the normal tissues had less than fivefold greater $\alpha 9$ -nAChR expression than the paired tumor tissues (**Fig. R2-3a**, bars 5 and 6). We next categorized each tumor vs. normal tissue pair according to the clinical stage of the tumor (**Fig. R2-3b**). Advanced-stage tumors were associated with substantially higher levels of $\alpha 9$ -nAChR mRNA expression. Data are presented as comparisons of the means of the fold ratios between paired tumor vs normal tissues and compared at each stage with the fold change in $\alpha 9$ -nAChR mRNA expression levels in stage 0 (ductal carcinoma in situ) tumor vs normal paired tissues as follows: stage 0 = 1.0-fold, stage 1 = 1.14-fold, stage 2 = 3.51-fold, stage 3 = 6.66-fold, stage 4 = 18.88-fold.

Figure R2-1 | Detection of nAChR subunits by reverse transcription-polymerase chain reaction (RT-PCR) in normal and cancerous human breast cell lines.

(a) MCF-7, MDA-MB-231, AU-565, MDA-MB-453, and BT-483 are transformed human breast cancer cells; MCF-10A and HBL-100 are considered normal human breast cells. The expression profiles of nAChR subunits in human SAEC (small airway epithelial), NHBE (normal human bronchial epithelial), and H157 (lung cancer) cells were also examined as described previously²². (b) Relative mRNA expression of different nAChR subunits in normal and tumor human breast tissues isolated from 50 breast cancer patients. The cDNA was used for RT-PCR analysis, and the experiment was repeated twice. The percentage of occurrence is shown.

Figure R2-2 | Analysis of $\alpha 9$ -nicotinic acetylcholine receptor ($\alpha 9$ -nAChR) expression in human breast carcinoma tissues.

Determination of $\alpha 9$ -nAChR mRNA levels using reverse transcription-polymerase chain reaction (RT-PCR). Both $\alpha 9$ -nAChR and glucuronidase (GUS) transcripts were detected as single bands (403 and 165 base pairs, respectively) in both tumor and adjacent normal tissues. PCR was performed for 30 cycles. The agarose gel image showed 50 randomly chosen and representative patients.

Figure R2-3 | Paired tumor and normal tissue samples categorized according to the types and degree of $\alpha 9$ -nAChR mRNA expression differences.

(a) The levels of $\alpha 9$ -nAChR mRNA expression were subdivided into four groups depending on the extent of the difference in expression between tumor and normal tissue (<2, 2–5, 6–9, and >10-fold). The percentage of occurrences and the total number of tumor-normal pairs are presented for each category. (b) Relative expression of $\alpha 9$ -nAChR mRNA in tumor and normal tissue pairs grouped according to clinical breast cancer stage. The tumor-normal tissue pairs for which relative levels of $\alpha 9$ -nAChR mRNA were divided into five subgroups according to the clinical staging criteria recommended by the American Journal of Critical Care (AJCC). The data shown are the means of the fold ratios of expression in paired tumor and normal tissues. The error bars indicate 95% confidence intervals. The numbers of paired samples at each stage are indicated above the bars.

In this revised manuscript, we added and revised some statements as follows. Smoking-induced tumor progression, especially that of nicotine and tobacco-specific nitrosamine 4-(methylnitrosamino)-1-(3-pyridyl)-1-butanone (NNK) via CHRNA7, has been sufficiently investigated^{22,23}. As CHRNA7 is often associated with lung cancer²³, our previous studies^{20,21} have shown that CHRNA9 expression plays important roles in breast cancer progression. This study suggests that bupropion could have utility in smoking-related metastatic cancer patients with high nAChR expression. However, the target proteins (e.g., nAChRs) of bupropion to suppress nicotine-induced complex disassociation, downstream signals, and metastatic ability in BRCA remain to be fully elucidated. (Page 24, lines 3-10)

Additionally, to study the regulatory mechanism of bupropion on CHRNA9 and ERBB2, we selectively knocked down CHRNA9 (CHRNA9 KD) and ERBB2 (ERBB2 KD) expression using the clustered regularly interspaced short palindromic repeats/CRISPR-associated proteins (CRISPR/Cas9) system in BT474 and MDA-MB-231 cancer cells. (Page 21, lines 3-13; details in Supplementary Note 10)

6. Fifth, the relevance of nAChR9 on infectious diseases and cancers of viral origin is not clear. While there is a significant amount of literature on how nAChRs affect immune response, there is no concrete evidence that nAChRs facilitate HTLV-1 infection or contribute to hepatitis. Additional support should be provided to strengthen the data in Figure 5.

Reply:

In this revised manuscript, we added the **Supplementary Figure 27** and **Supplementary Note 9** to explore the relevance of CHRNA9 on the hepatitis B pathway in liver hepatocellular carcinoma (LIHC). (Page 16, lines 9-18; details in **Supplementary Note 9**)

7. Overall, studies described this manuscript are comprehensive and significant. It can be improved further by certain modifications, as suggested above.

Reply: Thank you for acknowledging our work.

We have carefully revised the manuscript according to all the suggestions from the reviewers. Our point-to-point responses are included as below. **Please note that all changes in the revised manuscript are marked red to facilitate the review.**

Reviewer #3:

Major comments:

1. The authors describe a novel approach to uncover the roles of membrane proteins as cancer hallmarks. The study is innovative and promising and also well-drafted.

Reply: Thank you for acknowledging our works.

2. The authors describe MP-community as MP community consists of an MP and its interacting proteins derived from the known and predicted PPIs. The term MP-community at the first glance it gives an impression that it consists exclusively of MPs, but actually in vivo MPs interact both with other MPs and with non-MPs; reflecting this, MP communities will most likely consist of both MP and non-MP proteins. Also, intracellular domains within the membrane-anchored proteins may translocate to cytoplasm or other cellular compartments after proteolytic processing; the translocated domains may then interact with other non-MP proteins. Can the authors clarify the makeup of the MP communities and the percentage of membrane-embedded proteins in each one of them? Is it feasible to refer to an MP-community as such if it's made up of largely non-MP proteins?

Reply:

Thank you for your comments. In this revised manuscript, we replaced the term “MP-community” to “MP partner community (MPP community)”, added the **Supplementary Figure 11** and **Supplementary Note 6**, and rewrote some statements to clarify the makeup of the MP communities. (Page 6, lines 17-21; details in **Supplementary Note 6**)

Please note that to identify the interacting proteins of each MP, we proposed a SIM strategy to calculate interacting scores (S_{SIM}) by selecting experimentally validated PPIs (called all PPI-based SIM) or direct physical PPIs (called direct PPI-based SIM) as PPI templates. To build the all PPI-based and direct PPI-based MPP communities, that is an MP with at least one binding partner, we integrated experimentally validated PPIs and direct PPIs with predicted PPIs identified by all PPI- and direct PPI-based SIM approaches, respectively.

3. Secondly, the authors have extracted 749,087 non-redundant PPIs from public repositories and compiled a set of PPIs for constructing MP-communities. However, they do not appear to have distinguished binary interactions from co-complex memberships in this set. In the absence of such distinction, the authors' inferences that the inferred networks display scale-free topology and their

data on the identification and properties of MP community hubs are untenable.

Reply:

In this revised manuscript, we distinguished binary interactions (i.e., direct physical interaction) from co-complex memberships to further present direct PPI-based SIM (Page 40, lines 5-13) and redrew the **Supplementary Figure 18** for analyzing topological properties of both all PPI- and direct PPI-based CaMPNets in human cancers. (Page 8, lines 20-23; Page 9 lines 15-16; details in **Supplementary Fig. 18**)

4. Materials and Methods: KEGG pathway set- The authors claim that 22 cancer pathways belonging to the categories ‘Cancers: Overview’ and 43 related pathways (e.g., cell cycle and adherens junction) were deemed cancer-related pathways. What were the criteria to select the “43 related pathways” as cancer-related pathways? This must be clarified in the text.

Reply:

We rewrote some statements for clarifying how to select the “43 related pathways” as cancer-related pathways in this revised manuscript. (Page 41, lines 8-12)

5. Page 6: ..cancer-specific CaMPNets... It is a little confusing to label the inferred CaMPNets as cancer-specific, unless the authors can establish that these CaMPNets are not observed in non-cancer datasets. Especially since a large number of cancer-related pathways that were included in the analysis participate in non-cancer physiological processes.

Reply:

In this revised manuscript, we deleted “cancer-specific” to avoid the misleading (Page 7, line 6).

6. CaMPNets are constructed based on the enrichment of co-expressed gene pairs between DEGs and specific cancer-related pathways. Have the authors examined if the co-expressed genes lie within close proximity to each other in the correlated pathways? This is because co-expression maybe less evident in the case signaling pathways that are often hierarchical in nature and as such the author’s approach may miss out on gene sets that belong to the same pathway but may not be sufficiently co-expressed.

Reply:

To examine whether the co-expressed genes lie within close proximity to each other in the correlated pathways, we evaluated and compared the shortest path lengths for the co-expressed gene pairs between MPP communities and cancer-related pathways and all the combinational gene

pairs within these pathways. Here, the genes of MPP communities that were recorded in the cancer-related pathways were used to measure the shortest path lengths of co-expressed gene pairs. For example, in the *CHRN9* community-regulated adherens junction (Fig. 5b), only four co-expressed gene pairs between the *EGFR* gene and differentially expressed genes (DEGs; e.g., *NECTIN3* and *WASF3*) could be measured as the shortest path lengths. The results show that the means of shortest path lengths (≤ 4.18) for all PPI-based and direct PPI-based co-expressed gene pairs across cancers (or in each cancer type) were significantly shorter than those (4.50, $P < 2.2 \times 10^{-16}$, Student's t-tests) for all gene pairs in 65 cancer-related pathways (Fig. R3-1).

Based on your comment and the above observation, we rewrote the statement in the discussion section for describing the limitation of our approach. (Page 24, line 16 to page 25, line 3)

Figure R3-1 | Shortest path length distributions of the co-expressed gene pairs (solid lines) of differentially expressed genes (DEGs) between all/direct PPI-based MPP communities and cancer-related pathways and all combinational gene pairs (black dot lines) between any two genes in cancer-related pathways.

Comparison of shortest path length distributions (a) across 15 cancers and (b) in each cancer type. To evaluate the shortest path lengths of co-expressed gene pairs between MPP communities and cancer-

related pathways, we assessed only the genes of MPP communities that were included in the cancer-related pathways.

Minor comments:

7. What were the criteria for selecting the 15 cancer types, why were only these cancers considered for the analysis?

Reply:

In this revised manuscript, we added a statement in **Supplementary Table 2** to describe how the 15 cancer types were selected. (details in **Supplementary Table 2**)

8. Page 4, Results “Identification of proteins interacting with MPs”- ...reflecting that many MP-interacting proteins still unknown. -> ...reflecting that many MP-interacting proteins are still unknown.

Reply:

Corrected. (Page 4, line 17)

9. Page 5, presented essential properties in humans-> performed/participated in essential properties in humans

Reply:

Corrected. (Page 5, line 25)

Reference:

1. Meyers, R.M., *et al.* Computational correction of copy number effect improves specificity of CRISPR-Cas9 essentiality screens in cancer cells. *Nat Genet* **49**, 1779-1784 (2017).
2. Xenarios, I., *et al.* DIP, the Database of Interacting Proteins: a research tool for studying cellular networks of protein interactions. *Nucleic Acids Res* **30**, 303-305 (2002).
3. Wang, H.Y., *et al.* beta-amyloid(1-42) binds to alpha 7 nicotinic acetylcholine receptor with high affinity - Implications for Alzheimer's disease pathology. *J Biol Chem* **275**, 5626-5632 (2000).
4. Jaldety, Y., *et al.* Sperm Epidermal Growth Factor Receptor (EGFR) Mediates alpha 7 Acetylcholine Receptor (AChR) Activation to Promote Fertilization. *J Biol Chem* **287**, 22328-22340 (2012).
5. Allen, C.M., Ely, C.M., Juaneza, M.A. & Parsons, S.J. Activation of Fyn tyrosine kinase upon secretagogue stimulation of bovine chromaffin cells. *J Neurosci Res* **44**, 421-429 (1996).
6. Swope, S.L. & Huganir, R.L. Binding of the Nicotinic Acetylcholine-Receptor to Sh2 Domains of Fyn and Fyk Protein-Tyrosine Kinases. *J Biol Chem* **269**, 29817-29824 (1994).
7. Kumar, P. & Meizel, S. Nicotinic acetylcholine receptor subunits and associated proteins in human sperm. *J Biol Chem* **280**, 25928-25935 (2005).
8. Charpantier, E., *et al.* alpha 7 neuronal nicotinic acetylcholine receptors are negatively regulated by tyrosine phosphorylation and Src-family kinases. *J Neurosci* **25**, 9836-9849 (2005).
9. Wang, K., *et al.* Regulation of the neuronal nicotinic acetylcholine receptor by Src family tyrosine kinases. *J Biol Chem* **279**, 8779-8786 (2004).
10. Szklarczyk, D., *et al.* The STRING database in 2017: quality-controlled protein-protein association networks, made broadly accessible. *Nucleic Acids Res* **45**, D362-D368 (2017).
11. Kotlyar, M., *et al.* In silico prediction of physical protein interactions and characterization of interactome orphans. *Nat Methods* **12**, 79-84 (2015).
12. Yu, H.Y., *et al.* Annotation transfer between genomes: Protein-protein interologs and protein-DNA regulogs. *Genome Res* **14**, 1107-1118 (2004).
13. Aranda, B., *et al.* The IntAct molecular interaction database in 2010. *Nucleic Acids Res* **38**, D525-D531 (2010).
14. Stark, C., *et al.* The BioGRID interaction database: 2011 update. *Nucleic Acids Res* **39**, D698-D704 (2011).
15. Mewes, H.W., *et al.* MIPS: analysis and annotation of genome information in 2007. *Nucleic Acids Res* **36**, D196-D201 (2008).
16. Ceol, A., *et al.* MINT, the molecular interaction database: 2009 update. *Nucleic Acids Res* **38**, D532-D539 (2010).

17. Kerrien, S., *et al.* Broadening the horizon - level 2.5 of the HUPO-PSI format for molecular interactions. *Bmc Biol* **5**, 44 (2007).
18. Wheeler, D.L., *et al.* Database resources of the National Center for Biotechnology Information: update. *Nucleic Acids Res* **32**, D35-D40 (2004).
19. Alanis-Lobato, G., Andrade-Navarro, M.A. & Schaefer, M.H. HIPPIE v2.0: enhancing meaningfulness and reliability of protein-protein interaction networks. *Nucleic Acids Res* **45**, D408-D414 (2017).
20. Lee, C.H., *et al.* Overexpression and activation of the alpha9-nicotinic receptor during tumorigenesis in human breast epithelial cells. *J Natl Cancer Inst* **102**, 1322-1335 (2010).
21. Wu, C.H., Lee, C.H. & Ho, Y.S. Nicotinic acetylcholine receptor-based blockade: applications of molecular targets for cancer therapy. *Clin Cancer Res* **17**, 3533-3541 (2011).
22. West, K.A., *et al.* Rapid Akt activation by nicotine and a tobacco carcinogen modulates the phenotype of normal human airway epithelial cells. *J Clin Invest* **111**, 81-90 (2003).
23. Schuller, H.M. Is cancer triggered by altered signalling of nicotinic acetylcholine receptors? *Nat Rev Cancer* **9**, 195-205 (2009).

Reviewers' comments:

Reviewer #1 (Remarks to the Author):

The authors have made major efforts to respond to concerns raised by previous reviewers. Here are some selected points that are important, but remain to be addressed, or to be clarified.

(i) The foundation of the described CaMPNets is the use of "749,087 experimentally validated PPIs" and "176,087 direct physical PPIs" as the PPI template for analysis. Therefore, it is essential to clearly describe the level of evidence that was used to define these PPIs as "experimentally validated", and the level of evidence used to define these PPIs as "direct physical PPIs". Simply citing a database is insufficient for this purpose. Critical evaluation of evidence for such designations for "experimentally validated" and "direct PPIs" is expected to add value to the current manuscript.

(ii) Related to Point #1, PPIs included in the cited database derived from high throughput screening experiments (e.g. MS based proteomics and yeast 2 hybrid screening) without secondary validation experimental data should not be considered as "experimentally validated", or "direct physical PPIs". For analysis, authors should explicitly state that the template includes both reported and validated PPIs. If the designation of "validated PPIs" will be used, authors may consider to clarify the definition and possibly to exclude these PPIs that have no validation data. To benefit readers, cautionary notes should be provided in the text (possibly in Methods and in Discussion).

(iii) The CHRNA9 PPIs were selected as an example for testing the CaMPNets generated hypothesis, which is necessary and very important. Concerns are associated with data quality.

a. Were these 18 partners all of predicted CHRNA9 partners in the CaMPNets?

b. For the co-IP and FRET validation assays, how were positive interactions defined (i.e. how was the cut-off defined)?

c. For Fig. S29, large variations were noticed for the co-IP validation experiments (Supplementary Fig. 29). The use of such data to support the validation rate as summarized in Fig. 6B is problematic. In addition, no controls for IP experiments were shown (Fig. S29). The validity of the labeled bands as indicated requires additional information, such as the predicted molecular weight and proper controls. Some bands for the same protein were mis-aligned, e.g. COPS6 (and CHRNA9), which raised questions which, or any, band represents COPS6. Evidence for specificity of each antibody used for immunoprecipitation should be provided (Table S7).

It is understandable that such endogenous co-IP experiments are challenging for some proteins. Therefore, proper controls and validated reagents are essential for the interpretation of the data.

It is also understandable that not all of predicted PPIs could be validated due to various limitations. Therefore, it is important to state limitations of the CaMPNets approach as reported.

The authors should add a section in the Discussion on limitations/challenges of the approach, including the definition of the PPI template, the interpretation of the CaMPNets data, and the need for experimental validation. With proper interpretation and stated limitations, the reported dataset may offer a valuable resource for the scientific community.

Minor points:

(v) Detailed description of the procedure used for the evaluation of the loss-of-function screens (Project Achilles data) will be helpful to readers (Supp. Note 4). The simple correlation between the CERES scores does not indicate a functional connectivity. What thresholds were used to distinguish between significant and not significant effect of individual gene knockout? If both binding partners pass the threshold, in how many cell lines? What positive control PPIs were analyzed to support the conclusions?

(vi) Since authors aim to link the predicted PPIs with the development of cancer, clinical data available from cancer patients may be used to support the functional connectivity between the predicted binding partners. One of the approaches that could be used for this purpose is the analysis of mutual exclusivity of genomic alterations.

(vii) "we assumed that MPs and their binding partners often have similar loss-of-function effects across multiple cell lines for each cancer type". It should be recognized that two proteins

that interact with each other may have opposite functions. Their function could also be context dependent.

Reviewer #2 (Remarks to the Author):

The revised manuscript by Lin et al., has expanded the original study by including a significant amount of additional supportive data. The interactions of CHRNA9 with 13 new interacting partners in breast cancer cells has been analyzed and discussed further, and they justify their findings in a rational fashion. It appears to me that this comprehensive analysis of the interactions of membrane proteins would be of interest, especially in the context of nAChR functions. The supportive data provided in the Response to Reviewers document is quite convincing as well.

Overall, I feel that this is a comprehensive piece of work that would be highly beneficial to those labs focusing on protein-protein interactions of membrane proteins.

Reviewer #3 (Remarks to the Author):

The authors have adequately addressed the concerns raised by me and I therefore, recommend the acceptance of the manuscript in its present form.

We have carefully revised the manuscript according to all the suggestions from the reviewers. Our point-to-point responses are included as below. **Please note that all changes in the revised manuscript are marked red to facilitate the review.**

Reviewer #1:

The authors have made major efforts to respond to concerns raised by previous reviewers. Here are some selected points that are important, but remain to be addressed, or to be clarified.

Major comments:

- (i) The foundation of the described CaMPNets is the use of “749,087 experimentally validated PPIs” and “176,087 direct physical PPIs” as the PPI template for analysis. Therefore, it is essential to clearly describe the level of evidence that was used to define these PPIs as “experimentally validated”, and the level of evidence used to define these PPIs as “direct physical PPIs”. Simply citing a database is insufficient for this purpose. Critical evaluation of evidence for such designations for “experimentally validated” and “direct PPIs” is expected to add value to the current manuscript.

Reply:

In this revised manuscript, we replaced the term “experimentally validated PPIs” with “reported PPIs”. We added to the Methods section a description of the levels of evidence that were used to define the reported PPI set and the direct PPI set. The description is based on information from the relevant databases (HUPO¹, DIP², IntAct³, PICKLE⁴, BioGRID⁵, and HIPPIE⁶) (Page 40, line 21 to page 42, line 5).

The reported PPI set comprises 749,087 non-redundant pairs collected from five databases, and the set of 31,810 direct PPIs (**Supplementary Data 7**) excludes the interactions that have no experimental validation data, no evidence of direct and physical binding, or that are only identified by one high-throughput screening experiment. Based on the definition of the direct PPI set, we now present reported PPI-based and direct PPI-based SIMs/CaMPNets in human cancers, and we modified the related figures and tables (**Fig. 3a, Supplementary Table 4, Supplementary Figs. 4, 5, 8, 9, 10, 11, 12, 13, 18, 19, 20, 23, 25, 43, and 44, and Supplementary Data 2, 4, 6, and 7**) and the relevant text. Please note that the observations and overall performances are substantially in accordance with our previous results.

- (ii) Related to Point #1, PPIs included in the cited database derived from high throughput screening experiments (e.g. MS based proteomics and yeast 2 hybrid screening) without secondary validation experimental data should not be considered as “experimentally validated”, or “direct physical PPIs”. For analysis, authors should explicitly state that the template includes both reported and validated PPIs.

If the designation of “validated PPIs” will be used, authors may consider to clarify the definition and possibly to exclude these PPIs that have no validation data. To benefit readers, cautionary notes should be provided in the text (possibly in Methods and in Discussion).

Reply:

In this revised manuscript, we replaced the term “experimentally validated PPIs” with “reported PPIs”. We also added a description of the levels of evidence that were used to define the reported PPI set and the direct PPI set as indicated in the reply to comment 1 (Page 40, line 21 to page 42, line 5). Because the reported PPI-based S_{SIM} approach could additionally identify the interacting proteins of 880 (34%) MPs in comparison to the direct PPI-based S_{SIM} approach (details in **Supplementary Note 2** and **Supplementary Fig. 5b**), we presented both reported PPI-based and direct PPI-based SIMs/CaMPNets in human cancers. In support of this pursuit, we added cautionary notes on the reported PPI set in the Discussion section and provided the associated PubMed identifier(s) and PSI-MI term(s) as supporting evidence for each reported PPI on the website (Page 24, line 22 to page 25, line 8).

- (iii) The CHRNA9 PPIs were selected as an example for testing the CaMPNets generated hypothesis, which is necessary and very important. Concerns are associated with data quality.
- Were these 18 partners all of predicted CHRNA9 partners in the CaMPNets?
 - For the co-IP and FRET validation assays, how were positive interactions defined (i.e. how was the cut-off defined)?
 - Fig. S29, large variations were noticed for the co-IP validation experiments (Supplementary Fig. 29). The use of such data to support the validation rate as summarized in Fig. 6B is problematic. In addition, no controls for IP experiments were shown (Fig. S29). The validity of the labeled bands as indicated requires additional information, such as the predicted molecular weight and proper controls. Some bands for the same protein were mis-aligned, e.g. COPS6 (and CHRNA9), which raised questions which, or any, band represents COPS6.
 - Evidence for specificity of each antibody used for immunoprecipitation should be provided (Table S7).

Reply:

- In this study, we identified 64 candidates with $S_{SIM} \geq 3.0$, including 14 candidates with $S_{SIM} \geq 3.6$, that could potentially interact with CHRNA9 and then hierarchically clustered them into 5 subgroups using similarity scores. Of these 64 candidates, 18 were selected for experimental validation of the method (details in **Supplementary Note 8** and **Supplementary Fig. 27a, b**). Thank you for understanding that such endogenous co-IP experiments are challenging for some proteins and that not all of the predicted PPIs could be validated due to various limitations. Due to limitations of budget and technology (e.g., no commercially available antibody), we tried

our best to select at least one candidate in each subgroup using the following criteria: (1) differentially expressed gene in certain cancer types (e.g., breast cancer); (2) high S_{SIM} ; (3) for each type of S score, including S_{jss} , S_{irs} , S_{es} , and S_{topo} , its score should be greater than the mean of all the candidates; and (4) a commercially available antibody.

- b. Please note that Supplementary Fig. 29 was changed to Supplementary Fig. 30 in this revised manuscript. To determine the positive interactions between CHRNA9 and 18 interacting candidates (or SLC16A7 and four interacting candidates; **Supplementary Fig. 32**) in the co-IP assay, we first measured the band intensities⁷⁻⁹ of immunoprecipitated proteins and input loading controls (**Supplementary Fig. 30c**) on blots using ImageJ software¹⁰. Next, the IP ratio (*IPR*) for each immunoprecipitated protein in comparison to its loading control was utilized to evaluate IP efficiency. Here, a candidate that passes the *IPR* threshold (>3%) of both reciprocal IP assays is considered as a positive interaction. For example, ABCB1 is a positive case in MDA-MB-231 cells (3.05% and 25.06%) but a negative case in A549 cells (2.63% and 1.61%; **Supplementary Fig. 30c** and **Supplementary Table 6**) (Page 52, line 20 to page 53, line 19). We found that the results of the IP analysis are similar to those of the FRET efficiency analysis.

To determine the positive interactions between CHRNA9 and 18 interacting candidates in the FRET assay, we measured the FRET background efficiency on BT474 and MDA-MB-231 cells by hybridization with the primary antibody of each interacting protein (without the CHRNA9 antibody), followed by secondary rhodamine and FITC dye conjugations. According to SIM strategy, CAV1 (caveolin-1) was not an interacting protein of CHRNA9; therefore, this protein pair was selected to measure the FRET efficiency as a negative control. Based on the background signals of 18 interacting candidates and the signal between CHRNA9 and CAV1 in both BT474 and MDA-MB-231 cells, the maximum FRET efficiency (i.e., ERBB4 in MDA-MB-231) with a standard error was selected as the threshold (0.045; dashed line) to define positive interactions (**Supplementary Fig. 31c-f**) (Page 54, lines 7-23)

In this revised manuscript, we also rewrote corresponding statements (Page 17, line 23 to page 18, line 13).

- c. Please note that Supplementary Fig. 29 was changed to Supplementary Fig. 30 in this revised manuscript. To validate the IP results in a reciprocal fashion, we further immunoprecipitated protein extracts with the CHRNA9 antibody, followed by western blotting using 18 interacting partner antibodies, whereas IgG antibody immunoprecipitation served as a negative control (**Supplementary Fig. 30a**). The molecular weight of each protein was labeled in the western blotting results (**Supplementary Fig. 30a, b**). IP results showing misalignment or large variation were repeated. For example, misalignment: COPS6 in MDA-MB-231, RT4, and MIA PaCa-2 cells; large variation: HCK in five cell lines and PLK1 in BT-474, MDA-MB-231, A549, and MIA PaCa-2 cells (**Supplementary Fig. 30b**).
- d. The specificity of each antibody used for immunoprecipitation was determined by three criteria. First, the IP result matches its molecular weight. Second, the IP result has the same molecular

weight to the endogenous protein in the loading control. Third, no unspecific binding was shown for the IgG antibody control (**Supplementary Fig. 30a**).

- (iv) It is understandable that such endogenous co-IP experiments are challenging for some proteins. Therefore, proper controls and validated reagents are essential for the interpretation of the data. It is also understandable that not all of predicted PPIs could be validated due to various limitations. Therefore, it is important to state limitations of the CaMPNets approach as reported. The authors should add a section in the Discussion on limitations/challenges of the approach, including the definition of the PPI template, the interpretation of the CaMPNets data, and the need for experimental validation. With proper interpretation and stated limitations, the reported dataset may offer a valuable resource for the scientific community.

Reply:

To validate the CHRNA9 interactions and quality of the antibodies, we immunoprecipitated protein extracts with CHRNA9 antibody and 18 protein candidate antibodies in a reciprocal fashion, whereas an appropriate IgG antibody was used as a negative control (**Supplementary Fig. 30a**). Using a strict criterion, we considered a candidate as having a positive interaction when it passes the threshold in both reciprocal IP assays.

In this revised manuscript, we added a section in the Discussion on limitations/challenges of the approach, including the definition of the PPI template, the interpretation of the CaMPNets data, and the need for experimental validation (Page 24, line 22 to page 25, line 8).

Minor comments:

- (v) Detailed description of the procedure used for the evaluation of the loss-of-function screens (Project Achilles data) will be helpful to readers (Supp. Note 4). The simple correlation between the CERES scores does not indicate a functional connectivity. What thresholds were used to distinguish between significant and not significant effect of individual gene knockout? If both binding partners pass the threshold, in how many cell lines? What positive control PPIs were analyzed to support the conclusions?

Reply:

In this revised manuscript, we created new versions of **Supplementary Figure 8** and **Supplementary Note 4** to present more clearly our analysis of functional connectivity (i.e., mutual exclusivity and co-occurrence) between the MPs and their positive/predicted binding partners using the experimental data of loss-of-function screens from the Project Achilles data¹¹ (Page 6, lines 2-16).

Additionally, the odds ratio and *P* value (Fisher's exact test) for determining the mutual exclusivity and co-occurrence of each MP PPI for the group of 155 cell lines and for cells grouped according to cancer type are provided on the website.

In this analysis, each gene with CERES scores ≤ -0.2 , -0.4 , -0.6 , -0.8 , or -1.0 (suggested threshold in the Project Achilles Data Portal¹¹) is respectively considered to have a significant gene-knockout effect in each cell line. The two genes of a PPI/protein pair must both either pass or not pass the given thresholds to be considered a co-occurrence of loss-of-function effects in a cell line (**Supplementary Fig. 8a, b**).

In comparison to the cell line numbers (8-26) of nine cancer types, we also observed that the median values of cell line numbers for the co-occurrence MP PPIs ($P < 0.05$, Fisher's exact test), whose both genes have significant gene-knockout effects, ranged from 1 to 8, indicating that co-occurrence MP PPIs were not frequently shared by multiple cell lines (**Supplementary Fig. 8c**).

We used 16,334 MP reported PPIs and 1,851 MP direct PPIs as the positive controls and determined their mutual exclusivity or co-occurrence of loss-of-function effects to compare with the MP predicted PPI sets and to support our conclusions. In summary, our results suggest that the two genes of the positive/predicted MP PPIs tend to exert co-occurrence of loss-of-function effects in most of the nine cancer types but only to a limited extent (details in **Supplementary Note 4**).

- (vi) Since authors aim to link the predicted PPIs with the development of cancer, clinical data available from cancer patients may be used to support the functional connectivity between the predicted binding partners. One of the approaches that could be used for this purpose is the analysis of mutual exclusivity of genomic alterations.

Reply:

In this revised manuscript, we added **Supplementary Figure 9** and expanded **Supplementary Note 4** to cover our analysis of the mutual exclusivity and co-occurrence of genomic alterations between the MPs and their positive/predicted binding partners using the mutation and copy-number alteration (CNA) data of TCGA from the cBioPortal¹² database (**Page 6, lines 2-16**).

Additionally, the odds ratio and P value (Fisher's exact test) for determining the mutual exclusivity and co-occurrence of each MP PPI across multiple samples in each cancer type are provided on the website.

- (vii) “we assumed that MPs and their binding partners often have similar loss-of-function effects across multiple cell lines for each cancer type”. It should be recognized that two proteins that interact with each other may have opposite functions. Their function could also be context dependent.

Reply:

We agree your comments, “two proteins that interact with each other may have opposite or similar functions, and their function could also be context dependent”. In this revised manuscript,

we rewrote the statements of **Supplementary Note 4** to clarify that two proteins that interact with each other may have similar (i.e., co-occurrence) or opposite (i.e., mutual exclusivity) functions.

Here, we asked whether MPs and their binding partners often exhibited mutual exclusivity or co-occurrence of loss-of-function effects in the group of 155 cell lines and in each type of cancer (**Supplementary Fig. 8a**). Additionally, we added **Supplementary Figure 9** to cover the analysis of the mutual exclusivity and co-occurrence of genomic alterations between the MPs and their positive/predicted binding partners using the mutation and copy-number alteration (CNA) data of TCGA from the cBioPortal¹² database (**Page 6, lines 2-16**). Our results suggest that the gene pairs of the MP positive/predicted PPIs tend to exhibit co-occurrence of loss-of-function effects (or genomic alterations) but only to a limited extent. (details in **Supplementary Note 4**).

Reference:

1. Kerrien, S., *et al.* Broadening the horizon - level 2.5 of the HUPO-PSI format for molecular interactions. *Bmc Biol* **5**, 44 (2007).
2. Xenarios, I., *et al.* DIP, the Database of Interacting Proteins: a research tool for studying cellular networks of protein interactions. *Nucleic Acids Res* **30**, 303-305 (2002).
3. Aranda, B., *et al.* The IntAct molecular interaction database in 2010. *Nucleic Acids Res* **38**, D525-D531 (2010).
4. Gioutlakis, A., Klapa, M.I. & Moschonas, N.K. PICKLE 2.0: A human protein-protein interaction meta-database employing data integration via genetic information ontology. *Plos One* **12**, e0186039 (2017).
5. Stark, C., *et al.* The BioGRID interaction database: 2011 update. *Nucleic Acids Res* **39**, D698-D704 (2011).
6. Alanis-Lobato, G., Andrade-Navarro, M.A. & Schaefer, M.H. HIPPIE v2.0: enhancing meaningfulness and reliability of protein-protein interaction networks. *Nucleic Acids Res* **45**, D408-D414 (2017).
7. Zhu, J., *et al.* RING finger protein 31 promotes p53 degradation in breast cancer cells. *Oncogene* **35**, 1955-1964 (2016).
8. Murrow, L., Malhotra, R. & Debnath, J. ATG12-ATG3 interacts with Alix to promote basal autophagic flux and late endosome function. *Nat Cell Biol* **17**, 300-310 (2015).
9. Peng, Y.C., *et al.* BRI1 and BAK1 interact with G proteins and regulate sugar-responsive growth and development in Arabidopsis. *Nat Commun* **9**, 1522 (2018).
10. Schneider, C.A., Rasband, W.S. & Eliceiri, K.W. NIH Image to ImageJ: 25 years of image analysis. *Nat Methods* **9**, 671-675 (2012).
11. Meyers, R.M., *et al.* Computational correction of copy number effect improves specificity of CRISPR-Cas9 essentiality screens in cancer cells. *Nat Genet* **49**, 1779-1784 (2017).
12. Cerami, E., *et al.* The cBio Cancer Genomics Portal: An Open Platform for Exploring Multidimensional Cancer Genomics Data. *Cancer Discov* **2**, 401-404 (2012).

REVIEWERS' COMMENTS:

Reviewer #1 (Remarks to the Author):

The authors have made a major effort to address main concerns raised by the reviewers. Cautionary notes included in the revised manuscript will allow readers to make critical evaluations of the dataset presented. With clarified definitions and stated levels of supporting evidence for the reported MP network, this manuscript is expected to offer a valuable resource for the scientific community.

REVIEWERS' COMMENTS:

Reviewer #1 (Remarks to the Author):

The authors have made a major effort to address main concerns raised by the reviewers. Cautionary notes included in the revised manuscript will allow readers to make critical evaluations of the dataset presented. With clarified definitions and stated levels of supporting evidence for the reported MP network, this manuscript is expected to offer a valuable resource for the scientific community.

Reply: Thank you for acknowledging our works.